Corrected: Publisher correction

# GWAS of bone size yields twelve loci that also affect height, BMD, osteoarthritis or fractures

Unnur Styrkarsdottir

Bone area is one measure of bone size that is easily derived from dual-energy X-ray absorptiometry (DXA) scans. In a GWA study of DXA bone area of the hip and lumbar spine ($N \geq 28,954$), we find thirteen independent association signals at twelve loci that replicate in samples of European and East Asian descent ($N = 13,608 – 21,277$). Eight DXA area loci associate with osteoarthritis, including rs143384 in *GDF5* and a missense variant in *COL11A1* (rs3753841). The strongest DXA area association is with rs11614913[T] in the microRNA *MIR196A2* gene that associates with lumbar spine area ($P = 2.3 \times 10^{-42}$, $\beta = -0.090$) and confers risk of hip fracture ($P = 1.0 \times 10^{-8}$, OR $= 1.11$). We demonstrate that the risk allele is less efficient in repressing miR-196a-5p target genes. We also show that the DXA area measure contributes to the risk of hip fracture independent of bone density.

#A full list of authors and their affiliations appears at the end of the paper.

Human height and other characteristics of the body's size and shape have been extensively studied in large genome wide association (GWA) studies, yielding many associated loci[1–8]. Studies of parameters of body composition, such as lean mass, fat mass, and bone mineral density (BMD) have also yielded a number of associated loci[9–19]. An important component of human height and body form is the size and shape of the bones. Not only is height directly related to bone size but the size and shape of bones may contribute to their strength and risk of fracture[20], and the tendency to develop osteoarthritis[21,22]. DXA scans are primarily performed to evaluate BMD and fracture risk. The scans assess the area ($cm^2$) and the bone mineral content (BMC, g) of the bone, to produce the BMD ($g/cm^2$). BMD from DXA scans has found over 100 loci in many GWA studies[11–17], and estimated BMD from quantitative ultrasound of the heel over 1000 independent associations in the large UK Biobank resource[18,19]. The few GWA studies of bone area of the hip or lumbar spine using DXA have not yielded significant loci[23–27], possibly due to small sample sizes. Recently, a study of hip shape models (HSM) derived from statistical shape modeling of DXA scans found eight loci associated with hip shape[28].

Here, we report a large GWA study of bone size using a simple parameter from DXA scans, the bone area. We find thirteen independent associations of common variants at twelve loci that replicate in population of European and East-Asian descent. These are the first genome wide significant loci for DXA bone area. We also investigate association of these variants with other bone related traits and diseases (BMD, height, osteoarthritis, and fractures), and find all variants but one to associate with one of these traits/diseases. We show that the relationship between DXA bone area and these traits is, however, complex. We find high genetic correlation between femoral neck area and hip fracture, and show that the DXA area measure contributes to the risk of hip fracture independent of bone density. We demonstrate that a variant in the microRNA *MIR196A2* gene at the *HOXC* locus, rs11614913, affects the efficiency of *MIR196A2* in repressing miR-196a-5p target genes.

## Results

**Genome wide association study and replication.** To search for sequence variants that contribute to bone size we performed a GWA study of bone area using DXA of the hip (total hip, and the sub-regions femoral neck, trochanter, and intertrochanteric/shaft, $N \geq 28,954$) and of the lumbar spine (L1–L4, $N = 29,059$) (Supplementary Fig. 1). We adjusted the bone area measurements for sex, age, height and body mass index (BMI) and tested for association between 33.4 million high quality sequence variants[29] and the DXA bone areas (Methods).

Sixteen variants, all common, at 14 loci satisfied our criteria of genome-wide significance[30] in the association analysis, and after conditional analysis across the loci (Methods); eight for the lumbar spine area, five for the total hip area, four for the intertrochanteric area, three for the trochanter area, and one for the femoral neck area (Table 1, Fig.1, Supplementary Fig. 2, Supplementary Data 1 and 2, Supplementary Table 1). Two of the hip area loci, at 17q24.3 near the *SOX9* gene and at 4q31.21 near the *HHIP* gene, have been reported to associate with hip shape[28]. The *SOX9* association is the same signal as reported for hip shape, whereas rs12507427 near the *HHIP* gene represents an independent signal at that locus.

We followed up these sixteen sequence variants in samples with up to 21,277 subjects of European and East Asian descent (Methods, Supplementary Table 2). All but three markers replicated significantly under a false discovery rate (FDR) of 0.05 (Table 1, Supplementary Data 3 and 4). The strongest

associations are rs11614913 in the microRNA gene *MIR196A2* with reduced lumbar spine area ($P = 2.3 \times 10^{-42}$, $\beta = -0.090$), rs143384 in the 5'UTR of *GDF5* with increase in both total hip ($P = 2.2 \times 10^{-22}$, $\beta = 0.071$) and trochanter area ($P = 1.1 \times 10^{-18}$, $\beta = 0.071$), rs12601029 at 17q24.3 near the *SOX9* gene with increased intertrochanteric area ($P = 6.2 \times 10^{-18}$, $\beta = 0.072$), and rs12507427 near *HHIP* for increase in femoral neck area ($P = 8.4 \times 10^{-14}$, $\beta = 0.054$).

We do not confirm the previously reported suggestive association with DXA hip area at the *PLCL1* gene[23] ($P \geq 0.70$), or lumbar spine area at the *HMGN3* gene[25] ($P \geq 0.56$), whereas our association of rs143384 in the 5'UTR of *GDF5* with area of both lumbar spine and hip is the same as the previous suggestive association with lumbar spine area[26] ($r^2 = 0.78$).

In addition to the *SOX9* and *HHIP* loci, we find association between DXA hip area measures and five of the eight recently reported hip shape loci[28] under FDR of 0.05 (Supplementary Data 5).

**Functional annotation of variants and enrichment analysis.** We carried out functional annotation for the lead variants, along with highly correlated variants ($r^2 > 0.8$), using available data for active chromatin marks in trait-relevant cells (Supplementary Data 6); histone H3 mono-methylation and tri-methylation at lysine K4 (H3K4me1 and H3K4me3, respectively) and histone H3 acetylation at lysine K27 (H3K27ac). This analysis identifies ten DXA area loci intersecting with H3K4me1 or H3K27ac marked regions in primary osteoblasts, indicative of enhancers. The target genes for these enhancers include *SOX9*, at 17q24.3 which is the key-regulator of chondrocyte differentiation[31], *HHIP* at 4q31.21 which encodes hedgehog interacting protein that is involved in an important signaling pathway in osteoarthritis[32], and *COL11A1*, a fibril forming collagen of the extracellular matrix (Supplementary Data 6). Six loci are found in H3K4me3 regions of which five localize proximal to a transcription start site, indicative of active promoters (Supplementary Data 6).

Enrichment analysis for gene functions using DEPICT highlights genes with predicted functions in ribonucleoprotein complex binding, as well as those in LSM5 and LSM7 protein–protein interaction subnetwork, implicated in splicing and mRNA decay in P-bodies[33], for lumbar spine area and hip trochanter area association results, respectively (Supplementary Data 7). DEPICT identified significant tissue enrichment (FDR < 0.05) for two out of the five DXA area phenotypes, i.e., for hip total area and hip trochanter area association results, in both cases involving cartilage tissue (Supplementary Data 8). Of note, we find chondrocytes and osteoblasts among the list of top fifteen enriched cell-types ranked according to nominal $P$ values in five and three of the five DXA phenotypes, respectively.

**Overexpression of MIR196A2 in HEK293T cells.** The strongest spine area associated SNP, rs1161491, is located in the microRNA *MIR196A2* gene at the *HOXC* gene cluster of developmental transcription factors. The T allele of rs11614913 introduces a wobble bond to the primary miR-196a hairpin transcript (U to G paring instead of C to G pairing) without affecting the seed sequences of either the 5p or 3p strands of the miR-196a duplex, and hence, the variant is not expected to influence recognition of target gene mRNAs. However, the variant might affect the efficiency of miR-196-5p target gene suppression either by affecting the thermodynamic stability[34] or processing of the mature miR-196-5p duplex[35–37] (Supplementary Note 1, Supplementary Fig. 3).

To test this we transfected HEK293T cells with *MIR196A2* expression plasmids containing either the C or T alleles of

**Table 1 GWS DXA bone area variants in the Icelandic discovery samples**

| Area | SNP (region) | Closest Gene | VA | EA / OA | Freq | Discovery samples | | Replication samples | | All sets combined | | |
|---|---|---|---|---|---|---|---|---|---|---|---|---|
| | | | | | | P | Effect | P | Effect | P | Effect (95% CI) | P het |
| LS | rs11614913 (12q13.13) | MIR196A2 | in pre-mir | T/C | 47.8 | $1.5 \times 10^{-23}$ | −0.094 | $1.5 \times 10^{-20}$ | −0.086 | $2.3 \times 10^{-42}$ | −0.09 (−0.103, −0.077) | $5.4 \times 10^{-1}$ |
| | rs143384 (20q11.22) | GDF5 | 5′UTR | G/A | 37.0 | $9.9 \times 10^{-17}$ | 0.080 | $3.5 \times 10^{-7}$ | 0.047 | $4.4 \times 10^{-21}$ | 0.063 (0.050, 0.076) | $1.3 \times 10^{-2}$ |
| | rs10917168 (1p36.12) | WNT4 | intergenic | T/A | 28.4 | $3.3 \times 10^{-12}$ | 0.072 | $2.6 \times 10^{-13}$ | 0.075 | $5.8 \times 10^{-24}$ | 0.074 (0.059, 0.088) | $8.4 \times 10^{-1}$ |
| | rs143793852 (18q21.1) | DYM | intron | C/CA | 43.4 | $5.6 \times 10^{-11}$ | 0.062 | $1.1 \times 10^{-3}$ | 0.036 | $1.2 \times 10^{-12}$ | 0.051 (0.037, 0.065) | $7.3 \times 10^{-2}$ |
| | rs8036748 (15q25.2) | ADAMTSL3 | intron | A/T | 46.2 | $1.3 \times 10^{-10}$ | −0.060 | $7.8 \times 10^{-5}$ | −0.037 | $2.2 \times 10^{-13}$ | −0.049 (−0.061, −0.036) | $8.2 \times 10^{-2}$ |
| | rs2585073 (15q25.2) | SH3GL3 | intron | C/G | 34.9 | $1.3 \times 10^{-10}$ | 0.063 | 0.55 | 0.006 | $5.6 \times 10^{-7}$ | 0.035 (0.021, 0.049) | $4.7 \times 10^{-5}$ |
| | rs9341808 (6q14.1) | BCKDHB | intron | C/A | 47.1 | $1.7 \times 10^{-10}$ | 0.060 | $8.9 \times 10^{-8}$ | 0.049 | $1.1 \times 10^{-16}$ | 0.054 (0.042, 0.067) | $4.0 \times 10^{-1}$ |
| | rs72979233 (11q13.4) | CHRDL2 | downstream | G/A | 25.7 | $4.4 \times 10^{-10}$ | −0.067 | $9.8 \times 10^{-4}$ | −0.034 | $2.1 \times 10^{-11}$ | −0.05 (-0.064, −0.035) | $1.2 \times 10^{-2}$ |
| Hip | rs143384 (20q11.22) | GDF5 | 5′UTR | G/A | 37.0 | $4.2 \times 10^{-18}$ | 0.085 | $1.0 \times 10^{-6}$ | 0.054 | $2.2 \times 10^{-22}$ | 0.071 (0.057, 0.086) | $3.6 \times 10^{-2}$ |
| | rs3753841 (1p21.1) | COL11A1 | p.P1284L | G/A | 39.1 | $1.0 \times 10^{-17}$ | 0.083 | $1.4 \times 10^{-5}$ | 0.048 | $1.3 \times 10^{-20}$ | 0.068 (0.054, 0.082) | $1.7 \times 10^{-2}$ |
| | rs9830173 (3p14.3) | ERC2 | intron | C/G | 38.4 | $6.0 \times 10^{-13}$ | 0.070 | $2.2 \times 10^{-3}$ | 0.035 | $8.0 \times 10^{-14}$ | 0.055 (0.041, 0.07) | $2.0 \times 10^{-2}$ |
| | rs1507462 (18q23) | | intergenic | A/G | 30.4 | $1.8 \times 10^{-12}$ | −0.073 | $1.5 \times 10^{-3}$ | −0.034 | $3.5 \times 10^{-13}$ | −0.054 (−0.069, −0.04) | $8.9 \times 10^{-3}$ |
| | rs72834687 (17q23.2) | TBX4 | intron | A/G | 24.2 | $1.7 \times 10^{-10}$ | −0.068 | $9.4 \times 10^{-2}$ | −0.021 | $2.6 \times 10^{-9}$ | −0.048 (−0.064, −0.032) | $4.3 \times 10^{-3}$ |
| Inter | rs12601029 (17q24.3) | SOX9 | intergenic | A/G | 33.5 | $4.4 \times 10^{-14}$ | 0.074 | $2.8 \times 10^{-5}$ | 0.068 | $6.2 \times 10^{-18}$ | 0.072 (0.056, 0.089) | $7.5 \times 10^{-1}$ |
| | rs1159421 (17q24.3) | SOX9 | intergenic | T/C | 44.7 | $1.6 \times 10^{-13}$ | -0.069 | $5.4 \times 10^{-3}$ | −0.051 | $4.6 \times 10^{-15}$ | −0.065 (−0.082, −0.049) | $3.8 \times 10^{-1}$ |
| | rs3753841 (1p21.1) | COL11A1 | p.P1284L | G/A | 39.1 | $2.2 \times 10^{-11}$ | 0.064 | $3.6 \times 10^{-3}$ | 0.041 | $7.4 \times 10^{-13}$ | 0.057 (0.041, 0.072) | $1.8 \times 10^{-1}$ |
| | rs9830173 (3p14.3) | ERC2 | intron | C/G | 38.4 | $7.9 \times 10^{-11}$ | 0.062 | $3.0 \times 10^{-2}$ | 0.032 | $3.2 \times 10^{-11}$ | 0.053 (0.037, 0.069) | $1.2 \times 10^{-1}$ |
| Troch | rs143384 (20q11.22) | GDF5 | 5′UTR | G/A | 37.0 | $6.7 \times 10^{-18}$ | 0.083 | $2.6 \times 10^{-3}$ | 0.043 | $1.1 \times 10^{-18}$ | 0.071 (0.055, 0.086) | $2.0 \times 10^{-1}$ |
| | rs3753841 (1p21.1) | COL11A1 | p.P1284L | G/A | 39.1 | $8.8 \times 10^{-11}$ | 0.062 | $5.7 \times 10^{-8}$ | 0.077 | $4.0 \times 10^{-17}$ | 0.067 (0.051, 0.082) | $3.8 \times 10^{-2}$ |
| | rs10783854 (12q14.1) | CTDSP2 | upstream | T/C | 35.6 | $5.6 \times 10^{-10}$ | −0.060 | $2.4 \times 10^{-1}$ | -0.019 | $3.1 \times 10^{-9}$ | −0.049 (−0.066, −0.033) | $3.0 \times 10^{-2}$ |
| FN | rs12507427 (4q31.21) | HHIP | 5′UTR | A/T | 43.1 | $1.2 \times 10^{-10}$ | 0.059 | $1.1 \times 10^{-4}$ | 0.110 | $8.4 \times 10^{-14}$ | 0.054 (−0.036, −0.007) | $3.9 \times 10^{-1}$ |

Results are shown for the DXA area phenotypes that were GWS in the discovery analyses, results for all markers across all DXA phenotypes in the discovery samples are shown in Supplementary Data 1, results for all markers across all DXA area phenotypes combined in Supplementary Data 4, and in the replication samples split by European and Asian descent in Supplementary Data 3. Conditional analysis between the two variants in the SOX 9 gene is shown in Supplementary Table 1. Results are shown for the Icelandic discovery set, the replication sets combined, and all sets combined. Region refers to chromosomal location. EA designate the effect allele and OA the other allele. Freq. is the frequency of the effect allele in the Icelandic samples. Gene refers to the nearest gene and VA (variant annotation) to effect on transcript or protein. The estimated effects are expressed as standardized values (standard deviations above or below the population average) per copy of the SNP allele. P values are two sided and derived from a likelihood ratio test (Methods). P het is heterogeneity p value and is derived from a likelihood ratio test (Methods)
LS lumbar spine, Hip total hip, Inter intertrochanteric/shaft, Troch trochanter, FN femoral neck

rs11614913 and assessed, by RNA-sequencing, whether the two *MIR196A2* alleles displayed differential effect on mRNA expression (Methods, see Supplementary Note 1 for detailed description of this experiment and results). We first defined a set of seventeen high confidence miR-196a-5p direct target genes based on conservation scores for predicted binding sites in 3′UTRs of mRNA sequences using TargetScan v7.2 (Supplementary Fig. 4). Based on gene ontology analysis these direct target genes are highly overrepresented in anterior/posterior pattern specification (25-fold; $P = 1.7 \times 10^{-5}$), as well as in skeletal system development (13-fold; $P = 2.5 \times 10^{-5}$) (Supplementary Table 3). In the transfected HEK293T cells, 14 of the miR-196a-5p direct target genes are expressed (Supplementary Note 1). In response to overexpression of either alleles of *MIR196A2*, these 14 direct target genes are significantly down-regulated compared to the rest of the *MIR196A2* transfected HEK293 expressome ($P$ adjusted < 0.05) (Fig. 2, Supplementary Figs. 5 and 6, Supplementary Table 4). Contrasting the effects of the C and T alleles on suppression of the 14 target genes we observe a subtle difference, with the C allele suppressing more than the T allele (Fig. 2c, d). Permutation-based hypothesis test demonstrates that this suppression bias across the 14 target genes is statistically significant ($P$ value = 0.042) (Supplementary Note 1), showing that the rs11614913[T] allele of *MIR196A2* is less effective in repressing miR-196a-5p target genes. The subtle difference in the suppression of the target genes

between the two alleles of *MIR196A2* is to be expected given that the alleles are common with small effects on lumbar spine area and fracture risk. When analyzing gene enrichment for the entire set of repressed genes due to experimental induction of *MIR196A2* (C-allelles or T-alleles) expression, which represent both primary and secondary targets of miR-196a-5p, we find embryonic skeletal system morphogenesis among the significant enrichment of 31 functional gene ontology terms, supporting the relevance of *MIR196A2* for bone morphology (Supplementary Fig. 7).

One of the miR-196a-5p direct target genes is *HOXC8*, located in the same locus as rs11614913 in *MIR196A2*. The *HOXC* gene cluster of developmental transcription factors plays a critical role in limb development[38] and skeletal patterning[39]. In our adipose tissue samples from 746 individuals we find that the rs11614913 [C] allele correlates with 4.5% reduction in expression of the *HOXC8* gene ($P = 5.9 \times 10^{-9}$) (Supplementary Fig. 8), consistent with the effect of rs11614913 on the direct target genes in the *MIR196A2* transfected HEK293T cells. We note that another highly correlated variant ($r^2 = 0.90$), and potentially functional as it is in the promoter region of the *HOXC8* gene, rs371683123, also correlates with reduced expression of the *HOXC8* gene (Supplementary Fig 8, Supplementary Table 5). It is possible that rs371683123 and other correlated variants also contribute to *cis* regulation of *HOXC8* expression in addition to the effect of rs11614913 on *MIR196A2* function.

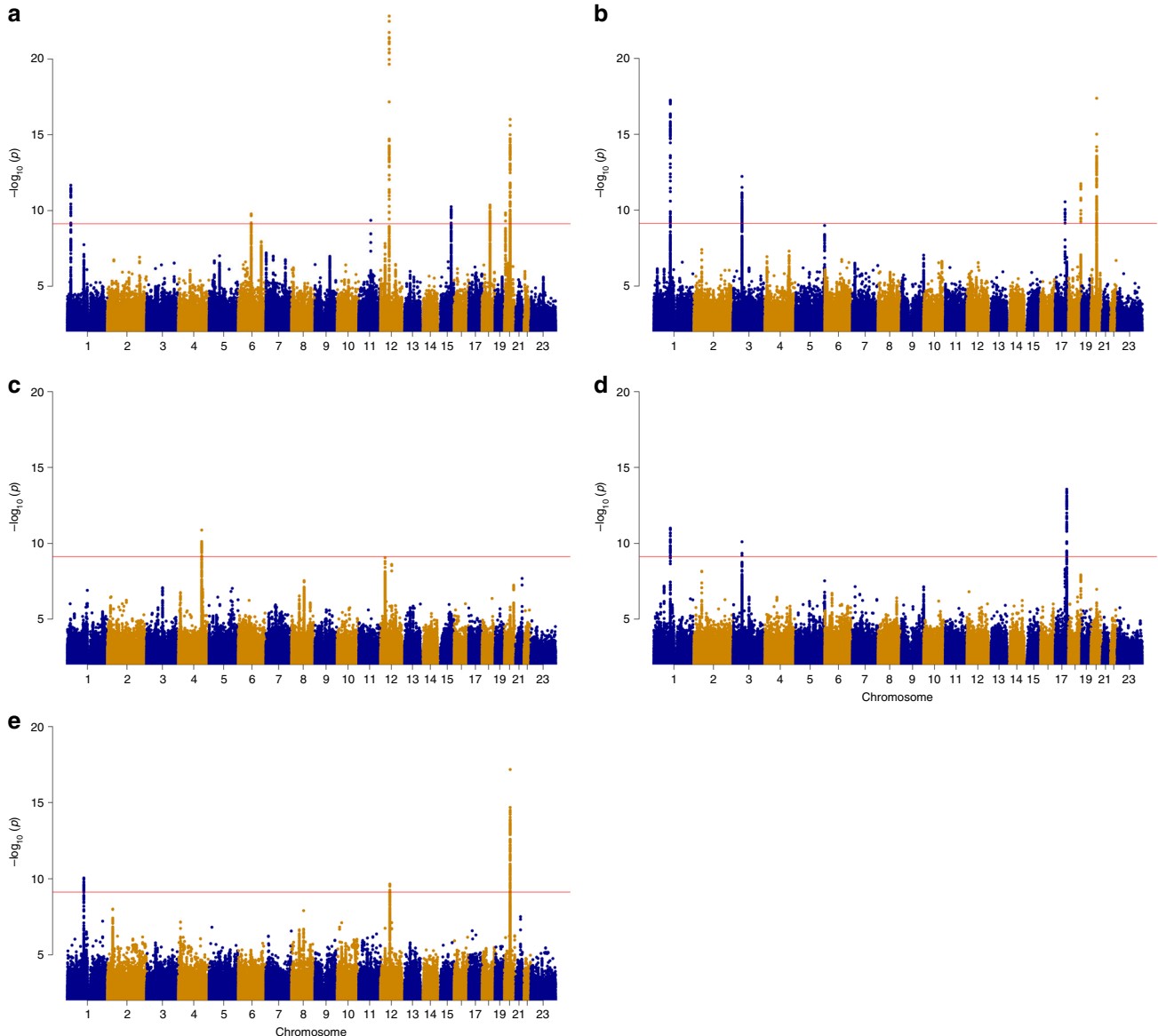

**Fig. 1** Manhattan plots of the genome-wide analysis of the DXA bone area measures. The *P* values (−log10) are plotted against their respective positions on each chromosome. Results are shown for all variants with significance level *P* < 0.001 and imputation information greater than 0.8. The red line denotes the significance level of intergenic variants, *P* ≤ 7.9 × 10−10. **a** Lumbar spine area (L1–L4) (*N* = 29,059), (**b**) Total hip area (*N* = 28,900), (**c**) Femoral neck area (*N* = 28,954), (**d**) Intertrochanteric/shaft area (*N* = 28,936), (**e**) Trochanter area (*N* = 28,944). Source data are provided as a Source Data file

**Association with other bone related phenotypes**. To determine whether the DXA bone area variants affect other bone phenotypes we examined association of the sixteen DXA area markers from our discovery analysis with osteoarthritis (*N*cases = 25,458–27,321) and vertebral (*N*cases = 3293) and hip fractures (*N*cases = 10,178) in European samples by either direct genotyping or in-silico look-up (Methods, Supplementary Table 2), and with height and BMD in public datasets (GIANT[1] and GEFOS[13], respectively).

Eight of the sixteen DXA bone area variants associate significantly with hip or knee osteoarthritis after correcting for FDR (0.05), four associate with hip fractures, six associate with BMD and twelve associate with height (Tables 2, 3). Two of the three variants that did not replicate significantly associate with height, at 15q25.2 and 12q14.1, and one, at 17q23.2, with hip osteoarthritis.

The osteoarthritis associating variants include the known knee osteoarthritis variant rs143384 in the *GDF5* gene[32,40] (*P* = 4.2 ×

$10^{-23}$, OR = 0.91), and the known missense variant in the *COL11A1* gene[32], rs3753841 (p.Pro1284Leu) that associates with hip osteoarthritis (*P* = 1.2 × 10$^{-11}$, OR = 0.92). Both variants associate with increased area and increased height (Tables 1, 3). Furthermore, we show that rs3753841[G] in *COL11A1* protects against hip dysplasia as shown by higher center-edge angle (β = 0.444, *P* = 3.4 × 10$^{-4}$), a measure of joint structure and shape related to dysplasia[41] (Supplementary Data 9).

The *MIR196A2* variant, rs11614913[T], that associates with reduced lumbar spine area (*P* = 2.3 × 10$^{-42}$, β = −0.090) also associates with increased risk of hip fractures (*P* = 1.0 × 10$^{-8}$ and OR = 1.11) (Table 2, Fig. 3), while it does not associate with DXA bone area measures of the hip (*P* ≥ 0.010) (Supplementary Data 1). This is the first sequence variant reported to associate with hip fractures to our knowledge. Furthermore, it associates with reduced lumbar spine BMD in the public GEFOS dataset[13] (Table 3), and confers modest risk of vertebral fractures in our study (*P* = 0.025, OR = 1.07) (Supplementary Table 6).

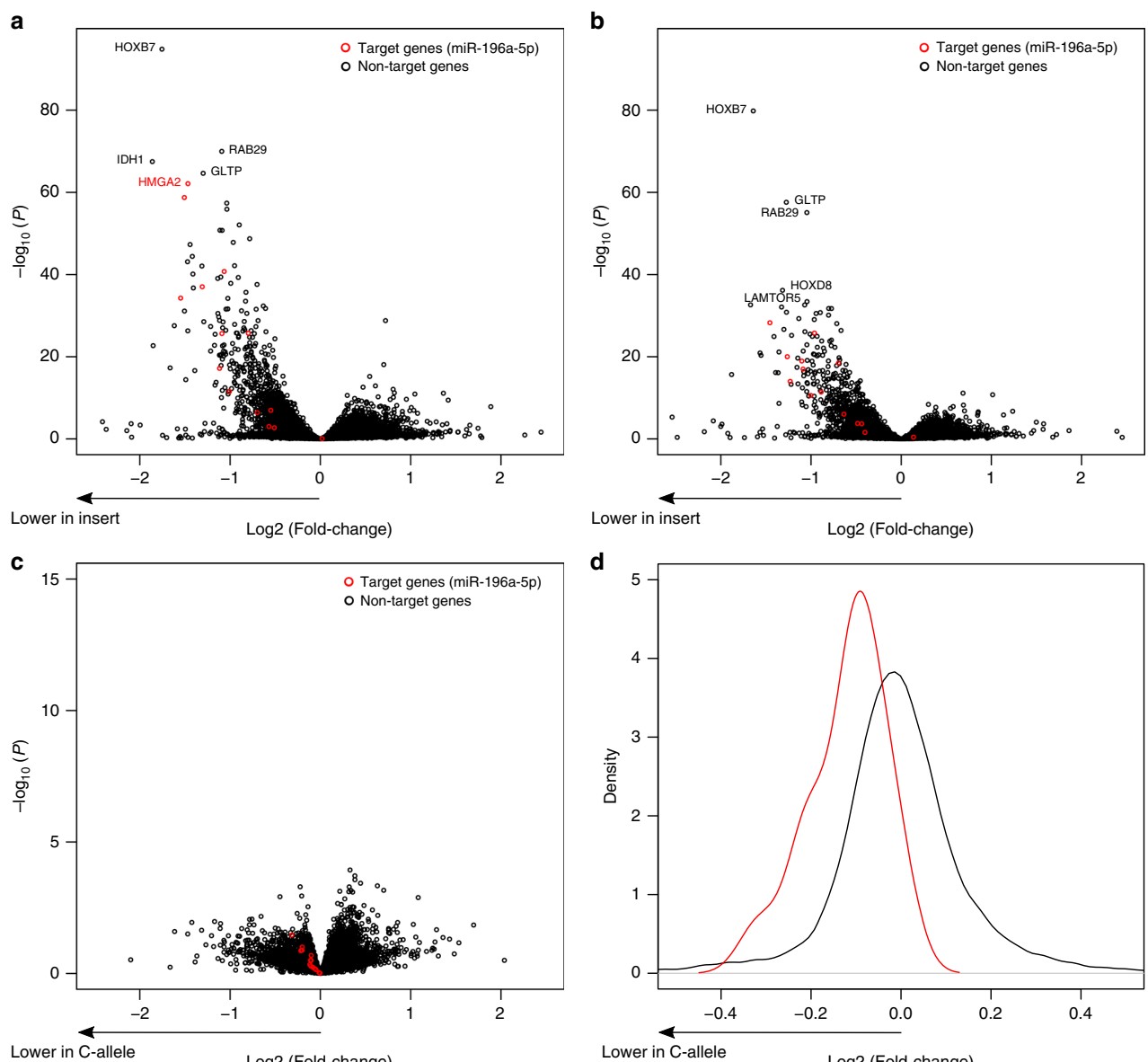

**Fig. 2** Repression of miR-196a-5p target genes by MIR196A2 C-allele or T-allele inserts. **a–c** Volcano plot shown for all pair-wise comparisons of cells with MIR196A2 C-allele or T-allele inserts and empty vector controls. In each comparison, red colored dots indicate the fourteen miR-196a-5p target genes. **a** C-allele compared with empty vector insert, (**b**) T-allele compared with empty vector insert, and (**c**) C-allele compared with T-allele. Negative log2-fold values reflect downregulation with miRNA insert and in **a** and **b**, whereas negative log2-fold values reflect downregulation of the C-allele in **c**. **d** Distribution of log2 fold-change values derived from the comparison of C-allele and T-allele transfected cells (negative log2 fold represents downregulation of the C-allele). The red and black curves represents the density distribution for the fourteen high-confidence miR-196a-5p target genes and other non-targets, respectively. The direction of effect is indicated below the x-axis by an arrow. Source data are provided as a Source Data file

Twelve of the DXA bone area variants[1] associate with height in the public GIANT dataset under a false discovery rate (FDR) of 0.05 (Table 3). This is despite initially correcting the DXA bone area measurements for height of the individual. Furthermore, of the 697 reported height variants[1], additional 24 associate with DXA bone area measures after correcting for the number of markers tested ($P$ Bonferroni $\leq 7.2 \times 10^{-5}$) (Supplementary Table 7). Although this high proportion of the DXA bone area variants associate with height we find little overall correlation between the effects of reported height[1] variants and their effect on DXA bone area measures in our data ($r^2 \leq 0.037$) (Supplementary Fig. 9). This is likely because some of the alleles that associate with decreased DXA bone area associate with decreased height, while other associate with increased height (Table 3, Fig. 4).

Mendelian randomization did not indicate a causal relationship between height and DXA bone area (Supplementary Table 8).

Six of the DXA bone area variants associate with BMD in the public GEFOS data under a false discovery rate (FDR) of 0.05 (Table 3), and additional eight reported BMD variants[13] associate with DXA area measures after correcting for the number of markers tested ($P$ Bonferroni $\leq 0.00071$) (Supplementary Table 7). We also observe some inconsistency in direction of effect on area and BMD (Table 3, Fig. 4), however, the overall correlation between the effects of reported BMD variants[11,13,15] and their effect on DXA bone area measures is positive for lumbar spine BMD ($r^2 = 0.21$, $P = 7.4 \times 10^{-5}$), whereas it is negative for hip BMD (femoral neck BMD) ($r^2 = 0.18$, $P = 0.00029$) (Supplementary Fig. 10). Mendelian randomization confirmed these

**Table 2 Association of DXA bone area variants with osteoarthritis and hip fractures**

| GWS Pheno | SNP (region) | EA / OA | Hip osteoarthritis (N = 21,458/601,367) | | | Knee osteoarthritis (N = 27,321/552,683) | | | Hip fractures (N = 10,178/703,740) | | |
|---|---|---|---|---|---|---|---|---|---|---|---|
| | | | P | OR | P het | P | OR | P het | P | OR | P het |
| LS | rs11614913 (12q13.13) | T/C | 0.45 | 0.99 | 0.014 | 0.021 | 1.02 | 0.18 | **$1.0 \times 10^{-8}$** | 1.11 | 0.034 |
| LS/Hip / Troch | rs143384 (20q11.22) | G/A | 0.99 | 1 | 0.36 | **$4.2 \times 10^{-23}$** | 0.91 | 0.87 | 0.60 | 1.01 | 0.48 |
| LS | rs10917168 (1p36.12) | T/A | **$4.4 \times 10^{-3}$** | 0.97 | 0.61 | 0.098 | 1.02 | 0.66 | 0.14 | 1.03 | 0.22 |
| LS | rs143793852 (18q21.1) | C/CA | 0.76 | 1.01 | 0.85 | 0.11 | 0.97 | 0.16 | 0.27 | 1.02 | 0.41 |
| LS | rs8036748 (15q25.2) | A/T | 0.25 | 1.01 | 0.84 | 0.063 | 1.02 | 0.66 | **$1.2 \times 10^{-2}$** | 1.05 | 0.69 |
| LS | rs2585073 (15q25.2) | C/G | 0.83 | 1 | 0.33 | 0.94 | 1 | 0.31 | 0.029 | 0.96 | 0.27 |
| LS | rs9341808 (6q14.1) | C/A | 0.044 | 0.98 | 0.64 | 0.091 | 0.98 | 0.61 | 0.17 | 1.02 | 0.17 |
| LS | rs72979233 (11q13.4) | G/A | 0.59 | 1.01 | 0.016 | **$2.4 \times 10^{-4}$** | 1.04 | $2.6 \times 10^{-3}$ | **$6.4 \times 10^{-3}$** | 0.95 | 0.92 |
| Hip/Inter/ Troch | rs3753841 (1p21.1) | G/A | **$6.7 \times 10^{-13}$** | 0.92 | 0.95 | 0.75 | 1 | 0.81 | **$6.5 \times 10^{-3}$** | 1.05 | 0.62 |
| Hip/Inter/ Troch | rs9830173 (3p14.3) | C/G | 0.086 | 0.98 | 0.17 | 0.24 | 0.99 | 0.97 | 0.17 | 1.03 | 0.33 |
| Hip | rs1507462 (18q23) | A/G | 0.69 | 0.99 | 0.83 | **$1.2 \times 10^{-3}$** | 0.97 | 0.43 | 0.61 | 1.01 | 0.061 |
| Hip | rs72834687 (17q23.2) | A/G | **$1.7 \times 10^{-4}$** | 1.05 | 0.7 | 0.29 | 1.01 | 0.39 | 0.82 | 1 | 0.46 |
| FN | rs12507427 (4q31.21) | A/T | 0.46 | 0.99 | 0.077 | 0.79 | 1 | 0.29 | 0.13 | 1.03 | 0.73 |
| Inter | rs12601029 (17q24.3) | A/G | 0.55 | 0.99 | 0.7 | **$1.3 \times 10^{-5}$** | 0.96 | 0.064 | 0.16 | 1.03 | 0.98 |
| Inter | rs1159421 (17q24.3) | T/C | 0.33 | 1.01 | 0.96 | **$2.8 \times 10^{-5}$** | 1.04 | 0.6 | 0.16 | 0.95 | 0.81 |
| Troch | rs10783854 (12q14.1) | T/C | 0.78 | 1 | 0.27 | 0.94 | 1 | 0.32 | 0.52 | 0.99 | 0.93 |

Results are shown for all sample-sets combined. GWS Pheno refers to the DXA bone area phenotype that was GWS in the discovery analyses. Region refers to chromosomal location. EA designate the effect allele and OA the other allele. N is the number of individuals in the analyses: cases/controls. P het is the heterogeneity P value and OR is odds ratio. P values in bold are significant for the phenotype under FDR of 0.05. P values are two sided and derived from likelihood ratio test (logistic regression)
LS lumbar spine, Hip total hip, Troch trochanter, Inter intertrochanteric/shaft, FN femoral neck

**Table 3 Association of DXA bone area markers with height and BMD**

| GWS Pheno | SNP (region) | EA/OA | Height[1] | | LS_BMD[13] | | FN_BMD[13] | |
|---|---|---|---|---|---|---|---|---|
| | | | P value | Effect | P value | effect | P value | Effect |
| LS | rs11614913 (12q13.13) | T/C | 0.72 | 0.001 | **$2.5 \times 10^{-11}$** | −0.059 | **$5.8 \times 10^{-7}$** | −0.038 |
| LS/Hip/Troch | rs143384 (20q11.22) | G/A | **$5.5 \times 10^{-120}$** | 0.08 | **$8.4 \times 10^{-3}$** | 0.024 | 0.2 | 0.01 |
| LS | rs10917168 (1p36.12) | T/A | **$5.7 \times 10^{-7}$** | 0.02 | **$2.6 \times 10^{-4}$** | −0.038 | **$8.7 \times 10^{-3}$** | −0.024 |
| LS | rs143793852 (18q21.1) | C/CA | **$1.7 \times 10^{-35}$** | 0.04 | 0.56 | 0.005 | 0.75 | 0.002 |
| LS | rs8036748 (15q25.2) | A/T | **$3.2 \times 10^{-53}$** | 0.048 | 0.12 | −0.015 | 0.016 | 0.02 |
| LS | rs2585073 (15q25.2) | C/G | **$4.5 \times 10^{-31}$** | −0.039 | 0.25 | 0.011 | 0.09 | −0.014 |
| LS | rs9341808 (6q14.1) | C/A | **$3.7 \times 10^{-4}$** | 0.026 | 0.87 | −0.001 | 1 | 0 |
| LS | rs72979233 (11q13.4) | G/A | **$5.7 \times 10^{-4}$** | −0.012 | **$1.6 \times 10^{-3}$** | −0.033 | 0.71 | −0.003 |
| Hip/Inter/Troch | rs3753841 (1p21.1) | G/A | **$3.1 \times 10^{-8}$** | 0.018 | 0.42 | −0.007 | 0.061 | −0.014 |
| Hip/Inter/Troch | rs9830173 (3p14.3) | C/G | 0.10 | −0.005 | 0.96 | −0.001 | 0.89 | −0.024 |
| Hip | rs1507462 (18q23) | A/G | 0.39 | −0.003 | 0.15 | 0.014 | 0.14 | 0.012 |
| Hip | rs72834687 (17q23.2) | A/G | 0.51 | 0.002 | 0.61 | 0.006 | 0.024 | 0.021 |
| FN | rs12507427 (4q31.21) | A/T | **$4.9 \times 10^{-63}$** | 0.053 | 0.41 | 0.007 | 0.24 | −0.009 |
| Inter | rs12601029 (17q24.3) | A/G | **$3.6 \times 10^{-6}$** | −0.015 | 0.97 | 0 | **$2.3 \times 10^{-3}$** | −0.025 |
| Inter | rs1159421 (17q24.3) | T/C | **$9.9 \times 10^{-5}$** | 0.012 | 0.46 | 0.007 | **$4.5 \times 10^{-6}$** | 0.035 |
| Troch | rs10783854 (12q14.1) | T/C | **$5.5 \times 10^{-10}$** | −0.025 | 0.64 | −0.005 | 0.51 | 0.006 |

Results are shown for height in the GIANT dataset[1] excluding the Icelandic samples, and the GEOFS dataset[13], that did not include the Icelandic samples, for lumbar spine (LS) and femoral neck (FN) bone mineral density (BMD). GWS Pheno refers to the DXA area phenotype that was GWS in the discovery analyses. Region refers to chromosomal location. EA designate the effect allele and OA the other allele. The estimated effects are expressed as standardized values (standard deviations above or below the population average) per copy of the SNP allele. P values in bold are significant for the phenotype under FDR of 0.05
LS lumbar spine, Hip total-hip, Troch trochanter, Inter intertrochanteric/shaft, FN femoral neck

associations (Supplementary Table 8). The negative correlation with hip BMD may reflect the periosteal expansion observed with age[42] that is not captured by age adjustments of the measures used. Likewise, the positive correlation between lumbar spine area and lumbar spine BMD may reflect vertebral degenerative changes that are not captured by these adjustments either. Differences in trabecular and cortical compartments of the bones that are not accounted for in the area or BMD measures may also explain these relationships[43].

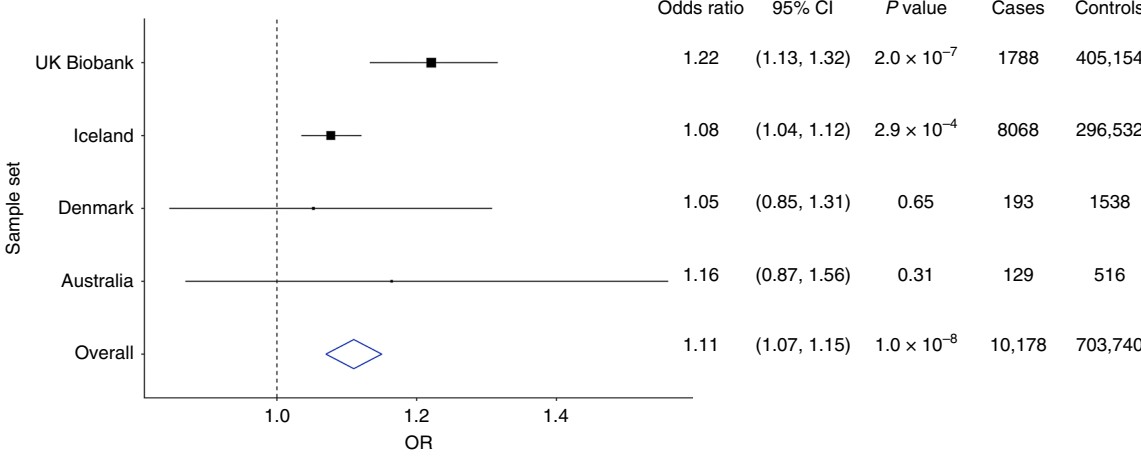

**Fig. 3** Forest plot showing estimated effect of rs11614913[T] in *MIR196A2* on hip fractures. The odds ratio, 95% confidence interval (CI), *P* value of association and number of cases and controls is shown for each study. Source data are provided as a Source Data file

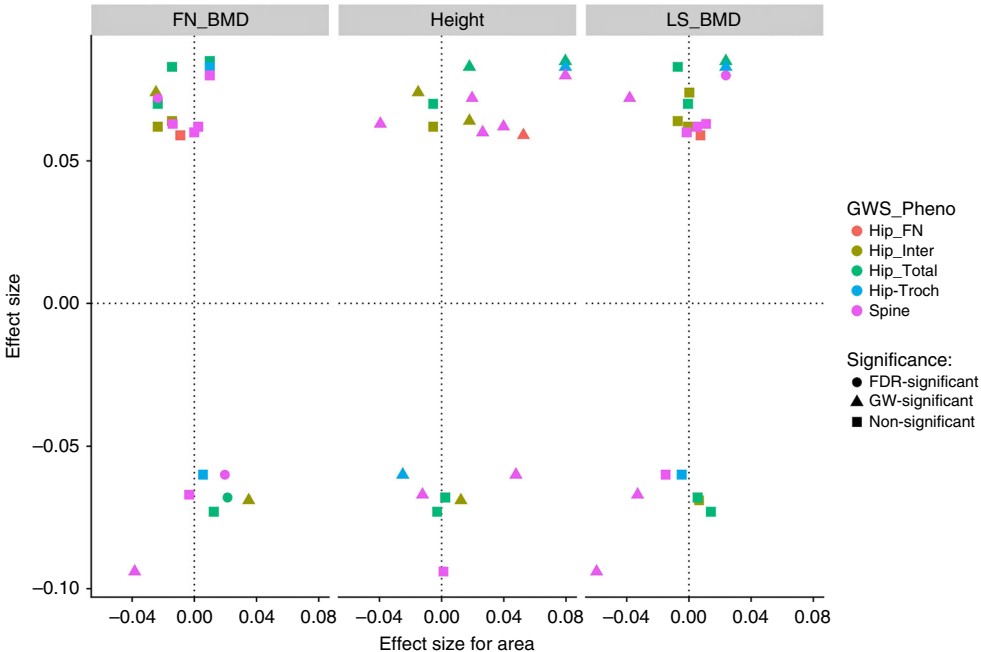

**Fig. 4** Effect on DXA bone area measures, height and bone mineral density. The effect on area measures are plotted on the *X*-axis and the effect on reported height variants from the GIANT consortium[1], and reported femoral neck BMD (FN_BMD) and lumbar spine BMD (LS_BMD) variants from the GEFOS consortium[13] are plotted on the *Y*-axis. Effects are in in standard deviations. The color of the dots/triangles/squares indicate the site of the DXA area measure, and the shape indicates the strength of association with height, and hip and lumbar spine BMD; squares, non-significant, dots, significant after correcting for FDR, and triangles, genome wide significant. Source data are provided as a Source Data file

**Genetic correlation between DXA area and related phenotypes.**
We examined the genetic correlation between the bone related traits and DXA bone area by cross trait LD score regression[44] across the Icelandic discovery set and the UK Biobank dataset (Table 4, Supplementary Table 9).

Consistent with the lack of correlation between the effect of variants on height and DXA bone area we find little genetic correlation between DXA bone area and height ($r_g = 0.064$ for lumbar spine, and $r_g = 0.14$ for hip). Likewise, we find consistent positive genetic correlation between DXA lumbar spine area and BMD ($r_g = 0.30$), and negative for hip BMD and hip area ($r_g = -0.15$).

For knee osteoarthritis there is no genetic correlation with any of the area measures ($r_g \leq 0.034$), whereas there is a weak positive

correlation between hip area and hip osteoarthritis ($r_g = 0.11$–0.18) but none for lumbar spine area and hip osteoarthritis ($r_g = 0.043$). The eight DXA bone area variants that associate with knee or hip osteoarthritis are both lumbar spine (3) and hip area (6) variants.

Hip fractures are substantially correlated genetically with the femoral neck area ($r_g = 0.76$), and even slightly more so than with femoral neck BMD in our data ($r_g = -0.57$). Recently, a large meta-analysis of fractures showed high genetic correlation between fracture and BMD ($r_g = -0.59$), and that BMD has a major causal effect on fracture[45]. Using multivariate logistic regression, we cannot differentiate between contribution through hip BMD, BMC, and the area measure on hip fractures, as each trait showed independent association with hip fractures while

**Table 4 Estimated genetic correlation**

| | | | | | Combined[a] | | |
|---|---|---|---|---|---|---|---|
| Iceland | UK Biobank | $r_g$ | s.e. | P value | $r_g$ | 95% CI | P value |
| LS area | Height | 0.056 | 0.038 | 0.15 | 0.064 | −0.008, 0.14 | 0.082 |
| Height | LS area | 0.16 | 0.13 | 0.22 | | | |
| Hip FN area | Height | 0.14 | 0.045 | 2.7E−03 | | | |
| LS area | LS BMD | 0.35 | 0.10 | 9.0E−04 | 0.30 | 0.13, 0.46 | 4.8E−04 |
| LS BMD | LS area | 0.20 | 0.15 | 0.17 | | | |
| Hip FN area | Hip FN BMD | −0.15 | 0.16 | 0.36 | | | |
| LS area | Knee OA | −0.040 | 0.059 | 0.50 | −0.034 | −0.15, 0.078 | 0.55 |
| Knee OA | LS area | 0.089 | 0.26 | 0.74 | | | |
| Hip FN area | Knee OA | −0.026 | 0.075 | 0.73 | | | |
| LS area | Hip OA | 0.076 | 0.077 | 0.32 | 0.043 | −0.10, 0.19 | 0.55 |
| Hip OA | LS area | −0.28 | 0.24 | 0.25 | | | |
| Hip FN area | Hip OA | 0.11 | 0.086 | 0.19 | | | |
| Hip fracture | FN BMD | −0.59 | 0.24 | 0.014 | −0.57 | −0.86, −0.28 | 9.8E−05 |
| FN BMD | Hip fracture | −0.56 | 0.19 | 2.5E−03 | | | |
| Hip FN area | Hip fracture | 0.76 | 0.28 | 6.6E−03 | | | |

We estimated the genetic correlation between the traits using cross-trait LD score regression[44] on the GWAS summary statistics from Iceland and UK Biobank. $r_g$ is the genetic correlation coefficient, s. e. is standard error, and CI confidence interval. Effect estimates obtained using weighted linear regression with s.e. and P values estimated with a block jackknife method[44]. Hip area measures are not available in the UK Biobank dataset limiting the analysis to hip area in Iceland versus the other phenotypes in the UK Biobank, and not vice versa. Full results are shown in Supplementary Table 8. LS lumbar spine, FN femoral neck, OA osteoarthritis, BMD bone mineral density. [a] Combined values do not include data from 'Hip FN area' rows

adjusting for the other phenotypes (Table 5, Supplementary Table 10, Supplementary Note 2). This indicates that all three measures, area, BMD and BMC, influence the risk of hip fracture. Of note is that lumbar spine area is associated with hip fractures while adjusting for all the other traits ($P = 0.0056$), while lumbar spine BMD is not significant ($P = 0.15$).

## Discussion

We have found thirteen sequence variants that associate with simple measures of bone size; area measures from bone density DXA scans. The scans assess the area (cm$^2$) and the bone mineral content (g) of the bone to produce the BMD (g/cm$^2$), which is used in evaluation of osteoporosis and risk of fractures. We are here using a component of these commonly performed scans that has not been extensively utilized in GWA studies as the BMD has. Our study is the first to report GWS associations to this trait. By using this simple measure, we also find association with previously reported hip shape loci, derived from statistical modeling of the DXA images[28]. Hence, the DXA bone area associations are likely to capture differences not only in the size of the bones, but also, at least partly, in their shape.

Importantly, we found strong association of two of these DXA bone area variants with osteoarthritis and one with hip fractures, the first significant hip fracture locus reported to our knowledge. These findings are in line with the idea that the size and shape of bones influence their strength, and previous speculations that they would influence the tendency to develop osteoarthritis. We also show that the DXA bone area measure, as well as the BMC measure of the DXA scans, are additional risk factors for hip fracture independent of BMD. The BMD has recently been shown to have a major causal effect on fractures[45]. Hence, all three measures (BMD, BMC, and area) influence the risk of hip fracture. The high genetic correlation that we find between the femoral neck area and hip fractures, which occur mostly at the femoral neck, also reflects the influence of DXA bone area measure on hip fracture. Interestingly, the lumbar spine area also contributes to the risk of hip fractures, independent of BMD and area of the hip. The association of the *MIR196A2* variant with lumbar spine area and hip fractures, but not hip area reflects this.

**Table 5 Contribution of DXA bone area, BMC and BMD to the risk of hip fractures**

| Phenotype | OR | 95% CI | P value |
|---|---|---|---|
| LS_Area | 0.97 | 0.95, 0.99 | 0.0045 |
| LS_BMC | 1.06 | 1.01, 1.11 | 0.020 |
| LS_BMD | 0.97 | 0.93, 1.01 | 0.12 |
| Hip_Area | 1.14 | 1.10, 1.17 | 6.6E−16 |
| Hip_BMC | 0.79 | 0.74, 0.84 | 1.7E−14 |
| Hip_BMD | 1.18 | 1.12, 1.24 | 1.3E−09 |

We estimated the contribution of the different DXA measures on the risk of hip fractures by multivariate logistic regression. The P values are two-sided and were derived by likelihood ratio test, adjusting for all the other phenotypes. Source data are provided as a Source Data file LS lumbar spine, BMC bone mineral content, BMD bone mineral density, CI confidence interval, OR odds ratio

Despite the little genetic correlation between the DXA bone area measures and height, most of these DXA area variants also associate with height, although with inconsistent direction of effects. We have recently demonstrated comparable relationship between osteoarthritis and height, with the osteoarthritis risk alleles associating either with increased or decreased height[32], resulting in little genetic correlation and little overall correlation between the effects of variants on these two traits, similar to what we report here for height and DXA bone area. These results underscore the complex nature of bone longitudinal and radial growth, mineralization, modeling and remodeling[43]. The complex association pattern of the DXA area variants with height, BMD, osteoarthritis and fractures may reflect different aspects of bone biology, such as differences between longitudinal and radial growth, or between trabecular and cortical compartments of the bones, the mineralization process, modeling and remodeling, and the periosteal expansion observed with age.

Finally, we show through overexpression in HEK293T cell line that the variant in *MIR196A2*, that associates with both decreased lumbar spine area and increased risk of hip fracture, directly influences repression of miR-196a-5p target genes, with the fracture associating allele, rs11614913[T], being less effective than

the rs11614913[C] allele. Many of the miR-196a-5p target genes play roles in development of the skeletal system, such as *HOXC8*, located within the *HOXC* gene cluster proximal to *MIR196A2*, underscoring the biological relevance of miR-196a-5p to bone morphology. Together these data warrants further studies in bone relevant cells.

## Methods

**Iceland discovery population**. The Icelandic samples with area measurements have previously been described in detail[15]. The area measures are derived from densitometers at the Landspitali University Hospital's DXA clinic and the Research Service Center's Heilsurannsokn DXA clinic, both using Hologic DXA machines. Values at the hip (total hip, intertrochanteric, trochanteric and femoral neck) and at lumbar spine (L1-L4) were corrected for machine, and adjusted for age, height and body mass index and standardized in each gender separately to have a mean = 0 and standard deviation = 1.

Information on osteoarthritis (OA) were derived from a national Icelandic hip or knee arthroplasty registry and from Landspitali University Hospital electronic health records. Fracture assessment was through Landspitali University Hospital's electronic health records from 1999 to 2016, excluding high trauma fractures according to the NOMESCO Classification of External Causes of Injuries or based on a detailed questionnaire. We only included patients who were 40 years or older at the time of joint replacements and vertebral fractures, and older than 50 years at the time of a hip fracture.

All participants who donated samples gave informed consent and the National Bioethics Committee of Iceland approved the study (VSN-15-198) which was conducted in agreement with conditions issued by the Data Protection Authority of Iceland. Personal identities of the participant's data and biological samples were encrypted by a third-party system (Identity Protection System), approved and monitored by the Data Protection Authority.

**Replication populations**. The Danish samples are postmenopausal women in the age range 55–86 years, taking part in the Prospective Epidemiological Risk Factor (PERF study)[46]. Area measures are from DXA-measurement (Hologic QDR2000) at the hip (total hip, femoral neck and intertrochanteric) and lumbar spine. Osteoporotic fractures included self-reported low trauma fractures and vertebral fractures assessed by digital measurements of morphologic changes. Hip and knee joint replacements and OA information were derived from the Land-spatientregistret, 1996–2014. The study was approved by the Ethical Committee of Copenhagen County and was in accordance with the principles of the Helsinki.

The Swedish samples were from the Swedish Malmo Diet and Cancer (MDC) study of men and women living in the city of Malmo in Sweden who were born between 1923 and 1945 (men) or between 1923 and 1950 (women). The inclusion examination was performed during 1991–1996. The participants (*n* = 28,449) were followed until first OA surgery, emigration from Sweden, or death until December 31 2005. Information on knee and hip arthroplasty for OA and mortality were from the national Swedish Hospital Discharge Register and the Swedish Causes of Death Register. Cases were defined as those who were treated with knee or hip arthroplasty (421 and 551, respectively) during the follow-up time. Controls matched each arthroplasty case for age, gender and BMI. The MDC study was approved by the research ethical committee at Lund University the MDC study (LU 51–90). All participants signed a written informed consent.

The Australian samples were derived from the Dubbo Osteoporosis Epidemiology Study (DOES)[47], including subjects in the age range 60–99 years. All are of Caucasian ethnicity. Area was measured at the lumbar spine and the hip femoral neck by DXA (LUNAR DPX-L) and corrected for gender, age, height and weight. Osteoporotic fractures included low trauma fractures assessed by questionnaire and ascertained by reviewing all radiography reports. The St. Vincent's Ethics Review Committee (Sydney) approved the study, and all subjects gave written informed consent.

The Dutch samples are from the Rotterdam study, an ongoing population-based prospective cohort comprising 14,926 Dutch individuals aged 45 years and older[48]. After its initiation in 1990 (RSI, *N* = 7983) the cohort had two follow up waves in 2000 (RSII, *N* = 3011) and 2006 (RSIII, *N* = 3932). Follow-up examinations were scheduled every 3–5 years. The Rotterdam Study was approved by the Medical Ethics Committee of the Erasmus MC and by the Ministry of Health, Welfare, and Sport of the Netherlands, implementing the Wet Bevolkingsonderzoek: ERGO (Population Studies Act: Rotterdam Study). All participants provided written informed consent to participate in the study and to obtain information from their treating physicians. DXA measurements in RSI-4 (2002–2004) and RSIII-1 (2006–2008) were done using Ge-Lunar Prodigy bone densitometer while in RSII-3 (2011–2012) aBMD was measured using iDXA total body fan-beam densitometer. The DXA area measures were adjusted for age, sex, and first 4 genetic principal components, and corrected for genomic control. The OA definition and related structural phenotypes were defined based on radiographs of the hip, hand and knee joints. We have used the following radiographic measurements to create (semi)-quantitative endophenotypes for the hip, knee, hand, finger and thumb joints: JSN (0–3 scoring), JSW (mm), Osteophytes (0–3 scoring), and Kellgren-Lawrence(KL)-score (0–5). Using these measurements

we have defined the following structural OA phenotypes: Finger/Hand/Thumb/Knee/Hip JSN sum score, osteophyte sum score, KL sum score and Hip JSW. In addition, based on clustering of the data, we made an additional subdivision of the Knee joint in lateral and medial side (JSN medial/lateral and Osteophyte score medial/Lateral), and divided the hip joint into the acetabulum and femoral head. Joint space width (JSW) was assessed at pelvic radiographs in anteriorposterior position andmeasured in mm, along a radius from the center of the femoral head. Within the Rotterdam Study, a 0.5 mm graduated magnifying glass laid directly over the radiograph was used to measure the joint space width of the hip joints. Acetabular dysplasia was measured using the Center-Edge angle or also known as the Wiberg (CE-angle). The angle was measured using statistical shape model (SSM) software. A continuous phenotype was used for the CE-angle, because of the normal distribution of the measured angles. Since the CE-angle of the right hip and the left hip has a high correlation (Pearson correlation coefficient 0.68), only the CE-angle of the right hip was used in our GWAS.

There are two studies from the United Kingdom; The UK Biobank sample-set (http://www.ukbiobank.ac.uk) and the Arthritis Research UK Osteoarthritis Genetics (arcOGEN, http://www.arcogen.org.uk)/United Kingdom Household Longitudinal Study (UKHLS) analyses (https://www.understandingsociety.ac.uk). arcOGEN is a collection of unrelated, UK-based individuals of European ancestry with knee and/or hip osteoarthritis (OA) from the arcOGEN Consortium[49]. Cases were ascertained based on clinical evidence of disease to a level requiring joint replacement or radiographic evidence of disease (Kellgren–Lawrence grade ≥ 2). The arcOGEN study was ethically approved by appropriate review committees, and the prospective collections were approved by the National Research Ethics Service in the United Kingdom. All subjects in this study provided written, informed consent. The United Kingdom Household Longitudinal Study, also known as Understanding Society, is a longitudinal panel survey of 40,000 UK households (England, Scotland, Wales and Northern Ireland) representative of the UK population. Since 2009 participants are surveyed annually and contribute information relating to their socioeconomic circumstances, attitudes, and behavior via a computer assisted interview. The study includes phenotypical data for a representative sample of participants for a wide range of social and economic indicators and biological sample collection including biometric, physiological, biochemical, and hematological measurements, as well as self-reported medical history and medication use. The United Kingdom Household Longitudinal Study has been approved by the University of Essex Ethics Committee and informed consent was obtained from every participant.

The UK Biobank study is a large prospective cohort study of ~500,000 individuals from across the United Kingdom, aged between 40–69 at recruitment[50]. DXA area measures were only available for the lumbar spine, for 5075 individuals. The measures were adjusted for age, height and body mass index in each sex separately and standardized to a mean = 0 and standard deviation = 1. Osteoarthritis at the hip and knee was defined as total hip or knee replacement after the age of 40 years and ICD10 codes indicating osteoarthritis. Femoral neck fractures comprised the hip fracture group and lumbar vertebrae fracture the vertebral fractured group. We only included individuals of UK European descent. UK Biobank's scientific protocol and operational procedures were reviewed and approved by the North West Research Ethics Committee (REC Reference Number: 06/MRE08/65), and informed consent was obtained from all participants. This research has been conducted using the UK Biobank Resource under Application Number 23359.

The Chinese Hong Kong samples are comprised of two samples of different sex, the Mr OS and Ms OS studies, aged 65 years and above[51]. The DXA bone area measures were derived using Hologic QDR-4500W densitometer and corrected for gender, age, height, and weight. All participants provided informed consent. The Clinical Research Ethics Committee of the Chinese University of Hong Kong approved the study.

The Korean samples are postmenopausal women who visited the Osteoporosis Clinic of Asan Medical Center (AMC, Seoul, Korea)[52]. Bone area was measured at the femoral neck and the lumbar spine (L1-L4) using Lunar-Expert XL, and Hologic-QDR 4500-A absorptiometers. The DXA area measures were corrected for machine, gender, age, height and weight. All participants provided informed consent. The AMC Ethics Review Committee (Seoul) approved the study.

The Vietnamese samples are from the Vietnam Osteoporosis Study (VOS), a population study in Ho Chi Minh City to map genome and exposome factors for predicting chronic diseases, including osteoporosis and fracture[53]. DXA bone area data was derived using Hologic Horizon densitometer and corrected for gender, age, height and weight. The study's procedure and protocol were approved by the research and ethics committee of the People's Hospital 115 on August 6, 2015 (Approval number 297/BVNCKH).

All participants in these studies provided informed consent, and we obtained approval from all institutional review board to carry out the study.

**DXA area measures**. Measurement of the different parameters is automated within the DXA machine software. Each machine is calibrated every morning before operation using a phantom. The two DXA machines used in Iceland were further calibrated by measuring a group of the same individuals on both machines. There are two main manufacturers of DXA machines, Hologic (https://www.hologic.com/hologic-products/breast-skeletal/horizon-dxa-system) and Lunar

(https://www.gehealthcare.com/en/products/bone-health-and-metabolic-health), each with their own software and means to optimize the results. All the measures for the Icelandic discovery samples were done using Hologic machines whereas the replication studies used either Hologic or the Lunar machines. There are some differences between the two machines, which are difficult to fully account for. However, using standardized measures rather than absolute values from the machines we believe most of those differences are accounted for in our analyses. Supplementary Fig. 1 was borrowed from the manufacturer of the Hologic DXA machines and illustrates the different areas measured. The Hologic manufacturers refer to the intertrochanteric/shaft area as intertrochanteric, and the Lunar manufacturers as shaft. This includes also the lesser trochanter. The femoral head is not included in the output from the DXA machines.

**Association analysis in the Icelandic samples.** We sequenced the whole genomes of 15,220 Icelanders using Illumina technology to a mean depth of at least $10 \times$ (median $32 \times$)[29,54]. SNPs and indels were identified and their genotypes called using joint calling with the Variant Effect Predictor (version 84)[55]. We improved genotype calls were by using information about haplotype sharing, taking advantage of the fact that all the sequenced individuals had also been chip-typed and long range phased. We then imputed the 33.4 million variants that passed high quality threshold into 151,677 Icelanders who had been genotyped with various Illumina SNP chips and their genotypes phased using long-range phasing. Using genealogic information, the sequence variants were imputed into un-typed relatives of the chip-typed to further increase the sample size for association analysis and increased the power to detect associations. All of the variants that were tested had imputation information over 0.8.

Quantitative traits were tested using a linear mixed model implemented in BOLT-LMM[56]. The area measures were corrected for age, height, and body mass index in each sex separately prior to regression analysis. The case-control analysis were done using logistic regression, adjusted for gender, age and county.

We applied the method of LD score regression[57] to account for inflation in test statistics due to cryptic relatedness and stratification, We regressed the χ2 statistics from our GWAS scan against LD score with a set of 1.1 M variants and used the intercept as a correction factor. We downloaded the LD scores from a LD score database (ftp://atguftp.mgh.harvard.edu/brendan/1k_eur_r2_hm3snps_se_weights. RDS; Accessed 23 June 2015). For the quantitative traits reported here, the estimated inflation factors based on LD score regression were 0.93 for the total hip area, 0.94 for the femoral neck area, 0.92 for the intertrochanteric area, 0.92 for the trochanteric area, and 0.93 for the lumbar spine area. For the case control analysis, the estimated inflation factors were 1.40 for hip osteoarthritis, 1.33 for knee osteoarthritis, 1.18 for hip fractures, and 1.07 for vertebral fractures.

We used the weighted Holm–Bonferroni method to allocate familywise error rate of 0.05 equally between five annotation-based classes of sequence variants[30]. This yielded significance thresholds of $2.6 \times 10^{-7}$ for high-impact variants (including stop-gained and loss, frameshift, splice acceptor or donor and initiator codon variants; $N = 8474$), $5.1 \times 10^{-8}$ for missense, splice-region variants and in-frame-indels ($N = 149,983$), $4.6 \times 10^{-9}$ for low-impact variants (including synonymous, 3′ and 5′ UTR, and upstream and downstream variants, $N = 2,283,889$), $2.3 \times 10^{-9}$ for deep intronic and intergenic variants in DNase I hypersensitivity sites (DHS) ($N = 3,913,058$) and $7.9 \times 10^{-10}$ for other non-DHS deep intronic and intergenic variants ($N = 26,108,039$).

**Genotyping of replication samples.** The samples from Denmark, Sweden, Australia, Hong Kong, Korea and Vietnam were directly genotyped on the Centaurus (Nanogen) single-SNP genotyping platform.

The Rotterdam Study cohorts (RS-I, RS-II, and RS-III) were genotyped using commercially available genotyping arrays, and genotyping quality control (QC) was done separately for each. The genome-wide arrays were used to impute variants with HRC[58]. The imputation was preformed using the Michigan Imputation server[59]. The server uses SHAPEIT2(v2.r790) to phase the data and Minimac 3 for imputation to the HRC reference panel (v1.0). Prior imputation the genotypes were filtered using scripts provided online (HRC Imputation preparation and checking: http://www.well.ox.ac.uk/~wrayner/tools/; 4.2.5). GWAS were carried out under an additive model in RV-test using HRC v1.0 imputations, with adjustment for age, sex and the first 4 principle components. EasyQC was used to conduct quality control across cohorts (excluded: MAF < 0.05). Cleaned results were combined in a joint meta-analysis (METAL: inverse variance weighing).

A total of 7410 arcOGEN cases were genotyped on the Illumina Human 610-Quad array, 670 arcOGEN cases were genotyped on the Illumina OmniExpress array, and 9296 United Kingdom Household Longitudinal Study (UKHLS) controls were genotyped on the Illumina CoreExome array. Prior to imputation all variants were mapped to GRCh37 and cases and controls were merged into a single dataset containing only those variants overlapping between the 3 datasets. Additional variants were excluded if they had a minor allele frequency (MAF) ≤ 1, were indels, were differentially missing between cases and controls (Fisher's exact test p < 0.0001). In addition, we performed a case-control analysis and visually inspected the cluster plots for any variant with p ≤ 5×10−8. Final quality control checks prior to imputation were performed using a HRC pre-imputation checking tool. Imputation was performed on 17,376 individuals and 126,188 variants using

the Michigan HRC imputation[58,59] server with Eagle2[60] for the prephasing. Post HRC-imputation we excluded related individuals by performing multi-dimensional scaling and identity by descent (using PI_HAT threshold > 0.2) in PLINK[61] using the directly typed variants. We performed association analysis using SNPtest[62] and we included the first ten principal components as covariates. There were 3312 total hip joint replacement cases and 2,421 total knee joint replacement cases and 9268 controls with 7,648,357 and 7,647,669 variants with an imputation information score > 0.3 and MAF > 1% for the hip and knee analysis, respectively.

The UK Biobank samples were genotyped was performed using a custom-made Affimetrix chip, UK BiLEVE Axiom[63], in the first 50,000 participants, and with Affimetrix UK Biobank Axiom array in the remaining participants;[64] 95% of the signals are on both chips. Wellcome Trust Centre for Human Genetics performed Iimputation by using a combination of 1000 Genomes phase 3, UK10K, and HRC reference panel reference panels in two steps; The HRC panel was used as first choice option, but for SNPs not in that reference panel the UK10K + 1000 Genomes panel was used[65]. We only used markers imputed based on the HRC panel due to problems in the UK10K + 1000 Genomes panel imputation for this study.

**Meta-analyses.** Results from the Icelandic discovery results and replication datasets were combined using an inverse-variance weighted meta-analysis, where different datasets we allowed to have different populations frequency for alleles and genotypes but assumed to have a common effect (fixed effect). Heterogeneity in the effect estimates was tested using a likelihood ratio test by comparing the null hypothesis of the effect being the same in both populations to the alternative hypothesis of either population having a different effect.

**Genetic correlations.** We used the cross-trait LD score regression method[44] and the summary statistics from the Icelandic and UK Biobank datasets for estimation of genetic correlation between pairs of traits. We used results for about 1.2 million variants, well imputed in both datasets, in this analysis, and for LD information, we used pre-computed LD scores for European populations (downloaded from https://data.broadinstitute.org/alkesgroup/LDSCORE/eur_w_ld_chr.tar.bz2). We calculated the genetic correlation between Icelandic GWAS summary statistic for one trait and the UK GWAS summary statistic for the other traits, and the vice versa, and then meta-analyzed those results to avoid bias due to overlapping samples. Area measures for the hip were not available for the UK Biobank dataset.

**Mendelian randomization.** Causal relationships between DXA bone area and height and BMD were estimated with a two sample Mendelian Randomization, using GIANT and GEFOS effect estimates for height and BMD, respectively, as instrumental variables. The Mendelian randomization was performed with the MendelianRandomization package in R[66].

**Functional annotation of DXA bone area associated variants.** For each lead variant we identified variants in linkage disequilibrium (LD) on the basis of in-house genotype data to the nearest 100,000 variants ($r^2 > 0.8$) to define an LD class. We then annotated these variants by intersection with chromatin immunoprecipitation (ChIP) signal data and Dnase hypersensitivity data for osteoblasts primary cells. The ChIP-seq data was derived from the ENCODE project[67] and downloaded in pre-processed (MACS v2 algorithm) bigWig format, representing analysis of acetylated histone H3 at lysine K27 (H3K27ac) marks with accession number ENCFF380JNO. To account for multiple hypothesis the signal P-values were adjusted by the Benjamini–Hochberg procedure and thresholded at the 1% FDR significance level. DNase hypersensitivity data (DHS) for osteoblasts was also downloaded in pre-processed narrow peak format (accession number ENCFF510LHV), and intersected with variant position in each LD class. Super-enhancer profile for osteoblast cells, based on H3K27ac data derived from the Encode project (GSM733739), was derived from Hnisz et. al.[68] Similar data for chondrocytes data was not available.

To identify enhancer targets genes, we made use of the Joint effect of multiple enhancers (JEME) resource[69]. Additionally, we used Genhancer elite predictions which is based on multiple data types (eQTLs, C-HiC, TF co-expression and eRNA co-expression, and distance) where the presence of at least two different types of data point to enhancer-gene targets. We permit all predictions in the elite version other than those where distance underlies the prediction, i.e., where distance is one of only two evidence levels. DEPICT analysis[70] was carried out using association variants with nominal P-value < 1e−5, separately for each of the five phenotypes.

**MIR196 experiments.** The full length mir196A2 cDNA (RefSeq: MI0000279) in pCMV-MIR mammalian expression vector was obtained from Origene in addition to an empty pCMV-MIR vector (product ID: pCMVMIR). Transformed TOP10 chemically competent cells (ThermoFisher C404006) were plated on LB plates containing 50 µg/ml kanamycin. Colonies were expanded in LB medium containing 50 µg/ml kanamycin. Plasmids were purified using Plasmid maxi kit (Qiagen 12163) following the manufacturer's protocol, resulting in Mir196A2_WT

and Mir_Empty plasmids. Sanger sequencing confirmed the sequence of Mir196A2_WT.

In order to create the Mir196A2 variant, we used the Q5 Site-directed mutagenesis kit (New England BioLabs E0554S) and the pCMV-mir196A2_WT plasmid was used as a template, with the following primers F-5′ CAAGAAACTGtCTGAGTTACATC ′3 and R-5′ TTGCCGAGTTCAAAACTC ′3. Sanger sequencing confirmed the sequence of Mir196A2_Mut.

Transfection experiments were independently done in six replicates for each condition ($n = 18$). One day prior to transfection, 500.000 HEK293T cells (ATCC® CRL-3216) were seeded into each well of a 6-well plate in 2 ml of Dulbecco's Modified Eagle Medium medium (ThermoFisher 31966021) supplemented with 10% fetal calf serum (ThermoFisher 10082147) and 50 units/ml penicillin and 50 μg/ml streptomycin (ThermoFisher 15070063). On the day of transfection, medium was replaced with 3 ml of the identical media as before without antibiotics. For each transfected well, 3.3 μg of Mir_Empty, Mir196A2_WT or Mir196A_Mut plasmids were diluted in 155 μl Opti-Mem (ThermoFisher 31985047) and 9.9 μl Fugene HD (Promega E2312). Incubated at room temperature for 10 min before 150 μl of transfection mix was added to the appropriate well.

Forty eight hours after transfection, cells were harvested for cell sorting. Cells were released from the plate surface with TrypLE Express (ThermoFisher 12605010) and then washed with 2% FBS (ThermoFisher 10500064) in dPBS (ThermoFisher 14190144). After washing, cells were diluted in 1 ml of 2% FBS in dPBS with 1:1000 DAPI (Sigma D9542). Live cells were then positively sorted for the expression of GFP using a Sony SH800S cell sorter, giving a purity of >85% transfected cells.

Cells were first gated by size with FSC Area/SSC Area and doublets were excluded by gating at FSC Height/FSC Area. Unstained cells were then used to specify the gating for negative cells. DAPI stained cells were also used to define the live/dead gate (Supplementary Fig. 11).

RNA from sorted cells was isolated using RNeasy plus mini kit (Qiagen 74136) following the manufacturer's protocol. After isolation, RNAsecure (ThermoFisher AM7006) was added to the RNA and samples heated for 10 min at 60 °C. Samples cooled down on ice and then placed in −80 °C.

The quality and quantity of total RNA samples were assessed and analyzed using the Total RNA 6000 Nano Chip for the Agilent 2100 Bioanalyser. TruSeq v2 RNA Sample Prep Kit (Illumina) was used to generate cDNA libraries derived from Poly-A mRNA. In brief, Poly-A mRNA from total RNA samples (0.5–1 μg input) was captured by hybridization to Poly-T beads. This was followed by fragmentation at 94 °C of the Poly-A mRNA, and first-strand cDNA prepared using SuperScript II Reverse Transcriptase (Invitrogen) and random hexamers, followed by second-strand cDNA synthesis, end repair, addition of a single A base, indexed adapter ligation, AMPure bead purification and PCR amplification. Bioanalyser was used to measure the resulting cDNA sequencing libraries using DNA 1000 Lab Chip. Sequencing was then carried out on a HiSeq2500 sequencer, with four indexed samples per lane. RTA (real-time analysis) and bcl2fastq software packages (Illumina), were used to generate basecalls and Fastq files, respectively. RNA transcript expression was then quantified with Kallisto[71] using cDNA sequences of the Ensembl v87 reference transcriptome[72]. Gene expression estimates were computed by aggregating transcript expression using tximport[73] R package. The gene expression difference in a pairwise comparison of groups; rs11614913[T], rs11614913[C] and empty was compared using DESeq2 R package[74]. Low expressed genes were excluded from the analysis if the average expression was below five fragments counts. P-values were corrected for multiple hypothesis testing using the Benjamini and Hochberg method.

**RNA sequencing and correlation in adipose tissue**. RNA sequencing analysis was performed on subcutaneous adipose tissue samples obtained from 749 Icelanders. RNA sequencing was as described above for the MIR196A2 experiment. Association between variant and gene expression was estimated using a generalized linear regression, assuming additive genetic effect and log-transformed gene expression estimates, adjusting for measurements of sequencing artefacts, demography variables, and hidden covariates[75].

**URLs**. For the arcOGEN Study, see http://www.arcogen.org.uk/; for the UK Biobank, see http://www.ukbiobank.ac.uk/; for the United Kingdom Household Longitudinal Study, see https://www.understandingsociety.ac.uk; for HRC pre-imputation checking tool, see http://www.well.ox.ac.uk/~wrayner/tools/#Checking; for rre-computed LD scores for European populations, see https://data.broadinstitute.org/alkesgroup/LDSCORE/eur_w_ld_chr.tar.bz2); for Hologic DXA machines, see https://www.hologic.com/hologic-products/breast-skeletal/horizon-dxa-system, and for Lunar DX machines, see https://www.gehealthcare.com/en/products/bone-health-and-metabolic-health.

## Data availability

The Icelandic population WGS data has been deposited at the European Variant Archive under accession code PRJEB15197, and the RNA sequencing results from the MIR196A2 experiments to Gene Expression Omnibus, accession code GSE128641. The GWAS summary statistics are available at https://www.decode.com/summarydata. The authors

declare that the data supporting the findings of this study are available within the article, its Supplementary Data files and upon request. A reporting summary for this Article is available as a Supplementary Information file. The source data underlying Figs. 1–4 in the main text, and for Supplementary Figs. 2, 4, 5, 6, 8, 9, 10, and for Table 5, are provided as Source Data files.

## Code availability

All custom codes used in this study are freely available online, in-house codes not applicable.

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

## Acknowledgements

We thank all the study subjects for their valuable participation, the staff from all studies and the participating physicians. A part of this research has been conducted using the UK Biobank Resource under Application Number 23359. arcOGEN (http://www.arcogen.org.uk/) was funded by a special purpose grant from Arthritis Research UK (grant 18030). Members of the arcOGEN Consortium are given in the Supplementary Note 3. Full acknowledgments are given in the Supplementary Note 4.

## Author contributions

U.S., D.G., U.T. and K.S. designed the study and interpreted the results. T.I., H.J., B.M., R.B. and G.S. provided phenotype data of the Icelandic subjects. K.T., C.G.B., L.S., J.C.S.L., N.L.S.T., T.C.Y.K., J.S.W.L., S.C.H., I.B., J.R.C., S.H.L., J-M.K., L.S.L., L.T.H-P., T.V.N., J.A.E., J.W., P-C.L., C.C., J.L., E.Z., F.R., J.v.M., A.U., I.J. and U.S. ascertained, recruited, managed and coordinated samples from non-Icelandic populations. K.G., A.B.A., O.A.S. and G.L.N performed and designed the experiments. U.S., G.T., S.H.L., F.Z., K.J., E.V.I., O.A.S., G.L.N., S.B., L.S., K.N., H.J., A.G., G.M. and D.F.G. analyzed the data. K.T. and C.G.B. analyzed the Dutch data. U.S., U.T., O.A.S., H.H., P.S., D.F.G. and K.S. drafted and edited the manuscript. All authors contributed to the final version of the manuscript.

## Additional information

**Competing interests:** The authors U.S, K.G., G.T., S.H.L., O.A.S., A.B.A., F.Z., K.J., E.V.I., H.J., A.G., G.L.N., S.B., L.S, K.N., P.S., G.M., I.J., H.H., U.T., D.G. and K.S. are employed by deCODE genetics/Amgen Inc. The remaining authors declare no competing interests.

Unnur Styrkarsdottir[1], Olafur A. Stefansson[1], Kristbjorg Gunnarsdottir[1], Gudmar Thorleifsson[1], Sigrun H. Lund[1,2], Lilja Stefansdottir[1], Kristinn Juliusson[1], Arna B. Agustsdottir[1], Florian Zink[1], Gisli H. Halldorsson[1], Erna V. Ivarsdottir[1], Stefania Benonisdottir[1], Hakon Jonsson[1], Arnaldur Gylfason[1], Kristjan Norland[1], Katerina Trajanoska[3,4], Cindy G. Boer[4], Lorraine Southam[5,6], Jason C.S. Leung[7], Nelson L.S. Tang[8,9], Timothy C.Y. Kwok[7,10], Jenny S.W. Lee[11,12], Suzanne C. Ho[13], Inger Byrjalsen[14], Jacqueline R. Center[15,16,17], Seung Hun Lee[18], Jung-Min Koh[18], L. Stefan Lohmander[19], Lan T. Ho-Pham[20], Tuan V. Nguyen[15,17,21], John A. Eisman[15,16,17,22], Jean Woo[11], Ping-C. Leung[7,23], John Loughlin[24], Eleftheria Zeggini[5,25], Claus Christiansen[14], Fernando Rivadeneira[3,4], Joyce van Meurs[4], Andre G. Uitterlinden[4], Brynjolfur Mogensen[2,26,27], Helgi Jonsson[2,28], Thorvaldur Ingvarsson[2,29,30], Gunnar Sigurdsson[2,31,32], Rafn Benediktsson[2,32], Patrick Sulem[1], Ingileif Jonsdottir[1,2,33], Gisli Masson[1], Hilma Holm[1], Gudmundur L. Norddahl[1], Unnur Thorsteinsdottir[1,2], Daniel F. Gudbjartsson[1,34] & Kari Stefansson[1,2]

[1]deCODE genetics/Amgen Inc., Reykjavik 101, Iceland. [2]Faculty of Medicine, University of Iceland, Reykjavik 101, Iceland. [3]Department of Epidemiology, ErasmusMC, 3015 GD Rotterdam, The Netherlands. [4]Department of Internal Medicine, ErasmusMC, 3015 GD Rotterdam, the Netherlands. [5]Wellcome Trust Sanger Institute, Hinxton CB10 1SA, UK. [6]Wellcome Trust Centre for Human Genetics, University of Oxford, Oxford OX3 7BN, UK. [7]Jockey Club Centre for Osteoporosis Care and Control, Faculty of Medicine, The Chinese University of Hong Kong, Hong Kong, China. [8]Faculty of Medicine, Department of Chemical Pathology and Laboratory for Genetics of Disease Susceptibility, Li Ka Shing Institute of Health Sciences,, The Chinese University of Hong Kong, Hong Kong, China. [9]CUHK Shenzhen Research Institute, Shenzhen 518000, China. [10]Department of Medicine and Therapeutics, Prince of Wales Hospital, Hong Kong, China. [11]Faculty of Medicine, Department of Medicine and Therapeutics, The Chinese University of Hong Kong, Hong Kong, China. [12]Department of Medicine, Alice Ho Miu Ling Nethersole Hospital and Tai Po Hospital, Hong Kong, China. [13]JC School of Public Health and Primary Care, Faculty of Medicine, The Chinese University of Hong Kong, Hong Kong, China. [14]Nordic Bioscience A/S, 2730 Herlev, Denmark. [15]Bone Biology Division, Garvan Institute of Medical Research, Sydney, NSW 2010, Australia. [16]School of Medicine Sydney, University of Notre Dame Australia, Sydney, NSW 2010, Australia. [17]St Vincent's Clinical School, University of New South Wales, Sydney, NSW 2010, Australia. [18]Division of Endocrinology and Metabolism, Asan Medical Center, University of Ulsan College of Medicine, Seoul 05505, Korea. [19]Orthopaedics, Department of Clinical Sciences Lund, Lund University, SE-22 100 Lund, Sweden. [20]Bone and Muscle Research Lab, Ton Duc Thang University, Ho Chi Minh City 700000, Vietnam. [21]School of Biomedical Engineering, University of Technology Sydney, Sydney, NSW 2007, Australia. [22]Clinical Translation and Advanced Education, Garvan Institute of Medical Research, Sydney, NSW 2010, Australia. [23]Institute of Chinese Medicine, The Chinese University of Hong Kong, Hong Kong, China. [24]Institute of Genetic Medicine, Newcastle University, Newcastle-upon-Tyne NE1 7RU, UK. [25]Institute of Translational Genomics, Helmholtz Zentrum München, 85764 München, Germany. [26]Department of Emergengy Medicine, Landspitali, The National University Hospital of Iceland, 101 Reykjavik, Iceland. [27]Research Institute in Emergency Medicine, Landspitali, The National University Hospital of Iceland, and University of Iceland, 101 Reykjavik, Iceland. [28]Department of Medicine, Landspitali–The National University Hospital of Iceland, 101 Reykjavik, Iceland. [29]Department of Orthopedic Surgery, Akureyri Hospital, 600 Akureyri, Iceland. [30]Institution of Health Science, University of Akureyri, 600 Akureyri, Iceland. [31]Research Service Center, Reykjavik 201, Iceland. [32]Department of Endocrinology and Metabolism, Landspitali, The National University Hospital of Iceland, 101 Reykjavik, Iceland. [33]Department of Immunology, Landspitali–The National University Hospital of Iceland, 101 Reykjavik, Iceland. [34]School of Engineering and Natural Sciences, University of Iceland, Reykjavik 107, Iceland

