## [Peer Review File · Nature Communications]

Reviewer #1 (Remarks to the Author):

This is a GWAS of bone area of the hip and spine performed by leaders in the field of osteoporosis genetics. Through large scale GWAS in the DECODE cohort followed by replication in other European and Asian datasets, the authors identify 12 loci associated with bone area. One of these was associated with hip fracture and six with osteoarthritis.

The study has been conducted competently and I have relatively minor suggestions.

Why are the genomic inflation factors so different between the different traits and why do some of these have very low values (0.82 for total hip area, 0.86 for femoral neck area) but others have very high values (1.40 for hip osteoarthritis, 1.33 for knee osteoarthritis)?

Were the logistic regression analyses adjusted for PCs?

Maybe the authors could report analyses with and without correction for height and/or bmi? It would be interesting to see the attenuation at some of these loci through conditioning on these factors and give some clues as to primary and secondary associations.

The authors could perhaps cite a recent GWAS by Trajanoska et al in BMJ on fracture. In this study the authors found that all the loci found for fracture were also BMD associated loci. Would the authors care to speculate whether there is evidence for fracture associated loci where the primary association is through bone area rather than BMD?

Bivariate LD score regression analysis looking at the genetic correlation between bone area and other anthropometric/disease/bone traits might be informative

p 5 GEFOS spelt incorrectly

Will the authors be making the GWAS summary results data available to the public?

Reviewer #2 (Remarks to the Author):

Nature Communications peer review report

Genome wide association study of bone size yields twelve loci that also affect height, bone density, osteoarthritis and fractures

Styrkarsdottir et al.

This article describes a GWAS in the Icelandic population of bone size at the hip, hip sub-areas, and the upper four lumbar vertebrae. Replication has taken place in a variety of cohorts, and attempts have been made to disentangle the relationship between bone size, height, BMD, and hip fractures and OA.

The work is novel, and will be of interest to the scientific community.

The paper overall is well-written, however it would benefit from some editing for clarity. In particular, the authors use bone area and bone size interchangeably, in the supplementary tables DXA size is used. One term should be used throughout. Furthermore, 16 variants at 14 loci were discovered, three failed to replicate ($16-3=13$), but the authors say that 12 loci were discovered – where this discrepancy lies is not obvious (this also relates to comment 4).

Major Comments:

1. Sub-phenotyping, especially in terms of sub-regions of the hip, is critical to the results of this study, but the way in which the bone areas were derived is not described (it may be automated within the software). Sup Fig 1 allegedly shows how the different parameters were derived, but in fact fails to do so. In particular:

- a. what determines the placement of the dotted yellow line box that defines the femoral neck?
- b. What determines the angle of the red solid line forming the border between the [greater] trochanter, and the “intertrochanteric” (this is not how the word is used clinically or anatomically) area?

Small changes in placement of these lines can have large implications on the size measurement ie. the phenotype of the GWAS.

2. Was the calculation of bone areas performed automatically by image recognition software, or was each scan read manually by an individual? Please provide references for the software, including estimates of intra- and inter-observer reliability. Different systems used in the different cohorts will no doubt give different results – what is the “inter-machine” correlation of these important parameters?

3. I note that the femoral head was excluded from analysis, but a major stream of analysis in the paper examines the overlap with hip OA – a disease formed between the femoral head and acetabulum. Why was the femoral head excluded? Can it be included?

4. It is not explicitly stated how the “loci” at SH3GL3 and ADAMTSL3 were separated. Was this on the basis of LD, distance, or both? No conditional analysis has been performed (or at least reported) at any locus. It is not uncommon for there to be more than one independent signal at a particular locus, so the analysis should be repeated conditioning on the top SNP to look for residual association at each locus. For example, the locus at POLD3 appears to cross a recombination peak, and may represent two signals.

5. In table 1, three markers did not replicate. These should be relegated to the status of “suspected” loci rather than “confirmed”, and the table, title, and discussion should be adjusted to reflect this.

6. It is suggested throughout the manuscript that the closest gene to the lead SNP is the one potentially responsible for the phenotype. There is a large literature on this often being untrue. It would be much better to perform a genome-wide gene-based test of association (eg MAGMA), and running gene-set enrichment analysis on the results – these analyses can be implemented in FUMA.

7. Line 131-134. Here it is suggested that rs3753841 associates with reduced osteophytes in knee and hand. Checking Sup table 7 reveals p values for these two associations of 0.89 and 0.52. Please correct this.

8. Line 140-142. Table 2. rs11614913 shows significant heterogeneity between studies when tested for association with hip fracture. Can we see a Forest plot to show this, and do the authors have any explanation why it may show opposite effects in different populations?

9. The associations between height, bone area, and BMD are indeed complex, and the authors have made efforts to disentangle this. Did they consider formal Mendelian Randomisation to look for a “causal” effect between the phenotypes?

Minor comments

Line 121 Extra closed bracket

Line 180 grammatically incorrect

Line 291 "OA" for "other allele" is mentioned in the legend but not shown in the table

Line 460 missing reference

Line 487 the UK Biobank corrected imputation was released a couple of months ago – could this not have been incorporated?

Supp Fig 2 – please colour all variants considered statistically significant in the discovery cohorts in a different colour (eg red) to aid their identification.

Table 2 and 3 – please highlight in bold the significant associations to make the table easier to read.

Reviewer #3 (Remarks to the Author):

The authors present interesting GWAS data and they aim at functionally evaluating the top GWAS variant that associates with lumbar spine area and hip fracture. The GWAS data are novel. I have some major comments re the functional exploration of the rs11614913 lead SNP and its associated variants. I recommend additional experimentation in trait-relevant cell types as well as clarification of presented results with respect to the pathology/biology (hip fracture/lumbar spine density)

Major comments

1. The hypothesis that rs11614913 might affect the thermostability of mature miRNA-196a duplex through introducing the wobble bond in miRNA duplex and thus alter the efficiency of target gene repression is plausible (not entirely novel). Please see <https://doi.org/10.3390/ijms18122529> - how do the authors comment on the results presented in this paper? Can the authors investigate the proposed candidate mechanism in more detail? E.g. testing efficiency of Ago loading of miRNA-196 on target suppression by luciferase reporter gene assays (co-transfection of miRNA_wt/mut overexpression plasmids with reporter vectors containing miRNA recognition sites); immunoprecipitation experiments with various flag-tagged human Ago proteins with subsequent detection of immuno-precipitated guide and passenger miRNAs in single stranded or duplex miRNA forms would give further insight, etc.

2. RNAseq in HEK cells (Fig. 1., Suppl Fig 5): It is not clear how the differences in the suppression of the identified miRNA target genes (by RNAseq in HEK cells) could functionally associate with hip fracture and lumbar spine bone density. Can the authors comment on the functionality of the target genes with respect to these traits? Are these genes expressed and affected by the genotype also in trait-relevant cells e.g. osteoblasts? Definition of a group of high-confidence miR-196a-5p targets is

not very convincing. It is not clear to me why the Pct score of exactly >0.86 was used. Did this score provide best results when referring to RNAseq analysis and data presentation (Fig. 1 and Suppl. Fig. 5.)? Can the RNA-seq data be analysed taking into account the miR-196a-5p targets that have been supported experimentally using the gold standard reporter gene assays (e.g. see DIANA-TarBase)? Several other transcripts seem altered by the overexpression of the variant-containing miRNA sequences in HEK cells. What are these targets? Taking 14 targets among these e.g. downregulated genes could yield similar plots as in Fig 1, Suppl. Fig 5 (based on predicted miR-196-5p mRNA targets) etc.

3. Other mechanisms beyond the effect of the genotype on miRNA 196-5p biology are possible. How were the lead SNP and its associated variants prioritized as most probably causal for the association with the trait, in particular for the rs11614913. The functional annotations, particularly for the rs11614913 and its associated variants, should be discussed beyond descriptive reporting in Suppl. Table 6, specifically in terms of potential functional variant prioritization. E.g, it could be that the promoter-located rs56154542 (the associated variant) rather than rs11614913 (the lead) determines HOXC8 levels. Can the authors test the associations of rs56154542 variation with HOXC8 expression? It is not clear to me why HOXC8 levels are reported in adipose tissue - the SNP effects can be cell-type specific and trait-relevant cells (e.g. osteoblasts) should be used to test the associations of the genotype with target gene expression (HOXC8 and other targets as determined by RNA-seq screen in HEK cells). Which of the affected targets in HEK cells (as defined by RNAseq – Fig 1) are expressed in e.g. osteoblasts? Do any of the miR-196-5p targets affect osteoblast biology.

Minor Comments:

- In comparison to empty control, T allele vector does not repress GAN while C allele vector does (rs11614913 lead SNP Suppl. Table 10). Can the authors comment on this?
- Please refer to lumbar spine (not spine only) are in abstract and text...
- rs11614913 lead SNP was shown to affect the expression levels of miR-196-5p (e.g. doi: 10.1038/srep19825) in seminal plasma. It would be important to measure the effects of the genotype on the expression of miR-196-5p in cells relevant for hip fracture/lumbar spine area traits, e.g. osteoblasts, if possible. Is the expression of miR-196-3p also affected?

Editor's comments:

We would specifically have to see improved annotation of variants, performed with current state-of-the-art computational methods, as mentioned by Reviewer #2, as well as presentation of the follow-up analyses in a more systematic way and with more consideration of the relevance to the studied trait, as pointed out by Reviewer #3.

Response: We are using the current state of the art computational methods to annotate the coding effects of variants, i.e. Variant Effect Predictor version 84. For functional annotation of non-coding sequence variants, we are making use of data resources from ENCODE, Roadmap and Fantom5 to identify potential consequences on genome regulation. We then make use of a resource developed for connecting enhancers to genes; this method is called the Joint Effect of Multiple Enhancers developed by Cao et al (Nature Genetics, 2017). In an effort for improvement, we now also include Genehancer predictions for enhancer-gene targets, see Supplementary Table 7. Additionally, for improvement, we use Depict developed by Pers et al (Nature Communications, 2015), as this method includes trait-relevant tissues, to assess enrichment for tissue types and gene functions; see summary of results in Supplementary Tables 8 & 9.

We have now added information and clarification to the MIR196A2 HEK293T experiment that was unclear to Reviewer #3, e.g. the conservation score threshold for selecting the primary target genes and possible functionality of the affected genes. We also include eQTL data for *HOXC8* (one of the target genes of MIR196A2), and a variant that is located in the promoter of *HOXC8*.

Regarding relevance of the adipocyte expression dataset to the studied trait. This dataset was used as we do not have access to whole genome genotypes and expression data from osteoblasts or chondrocytes, which are necessary to correlate expression and genotypes, and we are not aware of any publicly available dataset we could approach, apart from one that we have already contacted (see answer to Reviewer #3 - 3.). We are fully aware that regulation of expression between adipocytes and osteoblasts or chondrocytes may be different; however, we note that these cell types share origin from mesenchymal stem cells. We also note that bone is composed of other cell types that may be involved.

Please include a statement about data availability in your point-by-point letter accompanying your revisions. This would specifically apply to the deposition of GWAS summary statistics as requested by Reviewer #1.

Response: The GWAS summary data is available upon request. We did include this data availability statement in our initial submission: "The Icelandic population WGS data has been deposited at the European Variant Archive under accession code PRJEB15197. The authors declare that the data supporting the findings of this study are available within the article, its Supplementary Data files and upon request." We have now added "and the RNA sequencing results from the MIR196A2 experiments to Gene Expression Omnibus, accession code XXX" but this submission is in progress.

Reviewer #1 (Remarks to the Author):

This is a GWAS of bone area of the hip and spine performed by leaders in the field of osteoporosis genetics. Through large scale GWAS in the DECODE cohort followed by replication in other European and Asian datasets, the authors identify 12 loci associated with bone area. One of these was associated with hip fracture and six with osteoarthritis.

The study has been conducted competently and I have relatively minor suggestions.

Why are the genomic inflation factors to different between the different traits and why do some of these have very low values (0.82 for total hip area, 0.86 for femoral neck area) but others have very high values (1.40 for hip osteoarthritis, 1.33 for knee osteoarthritis)?

Response:

The reason for the difference between the magnitude of the inflation factors is that the case control analysis (e.g. osteoarthritis) was performed using logistic regression that does not account for relationship between individuals and therefore requires more adjustment than the quantitative trait regression which is based on mixed effects models that does account for relatedness between individuals. Unfortunately, we made an error in the scaling of the correction factors in the submitted manuscript which meant that the reported correction factor were reported to be smaller than the actually were. We have corrected this in the current manuscript and note that this error did not affect effect estimates or significance levels. The correct correction factors for total hip area and femoral neck area are 0.93 and 0.94, respectively.

Were the logistic regression analyses adjusted for PCs?

Response: Logistic regression analysis of the Icelandic data was adjusted for county of origin, of UKB data for 40 principle components, and for the Rotterdam data for the first 4 principle components. This information is included in the method section. The first few PC in Iceland capture similar information as the county of origin and do not add substantial adjustment to the association analysis.

Maybe the authors could report analyses with and without correction for height and/or bmi? It would be interesting to see the attenuation at some of these loci through conditioning on these factors and give some clues as to primary and secondary associations.

Response:

We have now run the analyses without adjustment for height and BMI (table below). The results are similar, and there is no significance difference in the effects between adjusted and unadjusted analyses (P_{het}).

Spine		P_{adj}	effect_{adj}	P_{unadj}	effect_{unadj}	P_{het}
rs11614913	MIR196A2	1.50E-23	-0.094	4.33E-23	-0.092	0.88
rs143384	GDF5	9.90E-17	0.08	3.65E-17	0.081	0.94
rs10917168	WNT4	3.30E-12	0.072	4.26E-13	0.074	0.89
rs143793852	DYM	5.60E-11	0.062	4.74E-11	0.062	1
rs8036748	ADAMTSL3	1.30E-10	-0.06	2.95E-10	-0.059	0.94
rs2585073	SH3GL3	1.30E-10	0.063	2.81E-10	0.062	0.94
rs9341808	BCKDHB	1.70E-10	0.06	1.12E-10	0.06	1
rs72979233	CHRD2	4.40E-10	-0.067	3.74E-10	-0.067	1
Total hip						
rs143384	GDF5	4.20E-18	0.085	3.34E-25	0.101	0.25
rs3753841	COL11A1	1.00E-17	0.083	6.16E-18	0.084	0.94
rs9830173	ERC2	6.00E-13	0.07	1.56E-13	0.072	0.88
rs1507462	.	1.80E-12	-0.073	1.97E-09	-0.062	0.45
rs72834687	TBX4	1.70E-10	-0.068	5.00E-09	-0.065	0.85
Inter						
rs12601029	SOX9	4.40E-14	0.074	2.73E-10	0.063	0.43
rs1159421	SOX9	1.60E-13	-0.069	1.40E-09	-0.057	0.37
rs3753841	COL11A1	2.20E-11	0.064	1.66E-12	0.068	0.77
rs9830173	ERC2	7.90E-11	0.062	8.72E-12	0.066	0.77
troch						
rs143384	GDF5	6.70E-18	0.083	3.27E-24	0.095	0.37
rs3753841	COL11A1	8.80E-11	0.062	6.01E-12	0.064	0.88
rs10783854	CTDSP2	5.60E-10	-0.06	1.65E-10	-0.06	1
FN						
rs12507427	HHIP	1.2E-10	0.059	7.94E-06	0.041	0.17

The authors could perhaps cite a recent GWAS by Trajanoska et al in BMJ on fracture. In this study the authors found that all the loci found for fracture were also BMD associated loci. Would the authors care to speculate whether there is evidence for fracture associated loci where the primary association is through bone area rather than BMD?

Response: We have now added citation to the GWAS by Trajanoska et al.

The hip fracture locus at 12q13.13 that we are reporting in this manuscript also associates with BMD as reported for GEFOS in Table 3. In the Icelandic dataset we also observe association with lumbar spine BMD but with significantly less effect than with area ($P_{\text{het}} = 0.000080$, LS-BMD: $P = 3.1E-07$, effect = -0.049 , and LS-area: $P = 1.5E-23$, effect = -0.094).

We performed a multivariable logistic regression for hip fractures, using age, height and sex adjusted measurements of DXA area, BMD and bone mineral content (BMC) of lumbar spine, as well as area and BMD of femoral neck, intertrochanteric and trochanteric and BMC of the total hip as explanatory variables. Each of these 10 phenotypes showed independent association with hip fractures, adjusting for the other 9 phenotypes, according to likelihood ratio test (see model output below).

*** model output.

Single term deletions

Model: hip_fractures ~ LS_area + hip_inter_area + hip_fn_area + hip_troch_area + LS_bmc + LS_bmd + hip_bmc + hip_fn_bmd + hip_inter_bmd + hip_troch_bmd

Phenotype	OR	95% CI	P value
LS_Area	0.97	0.95, 0.99	0.0063
LS_BMC	1.06	1.01, 1.11	0.027
LS_BMD	0.97	0.94, 1.01	0.16
Hip_Inter_Area	1.10	1.07, 1.13	2.9E-14
Hip_FN_Area	1.01	1.00, 1.01	4.9E-05
Hip_Troch_Area	1.04	1.03, 1.05	3.0E-19
Hip_BMC	0.83	0.79, 0.87	5.8E-14
Hip_FN_BMD	1.01	1.00, 1.02	0.013
Hip_Inter_BMD	1.09	1.06, 1.12	2.8E-08
Hip_Troch_BMD	1.04	1.02, 1.05	2.2E-08

The total hip is the sum of the inter, troch and fn, and therefore should not be included in the full model.

And with total hip instead of the sub-regions:

Phenotype	OR	95% CI	P value
LS_Area	0.97	0.95, 0.99	0.0045
LS_BMC	1.06	1.01, 1.11	0.020
LS_BMD	0.97	0.93, 1.01	0.12
Hip_Total_Area	1.14	1.10, 1.17	6.6E-16
Hip_Total_BMC	0.79	0.74, 0.84	1.7E-14
Hip_Total_BMD	1.18	1.12, 1.24	1.3E-09

Furthermore, five of the loci associated with bone area also nominally associated with hip fractures ($p < 0.05$). The figure below shows the estimated effect, with 95% confidence intervals, of each of these five variants for the listed phenotypes. These variants also show significant association with several BMD

or BMC measurements so we are not able to see whether the increased risk of fractures is due to bone area rather than BMD or BMC.

We have included these results in the manuscript as Supplementary material.

Bivariate LD score regression analysis looking at the genetic correlation between bone area and other anthropometric/disease/bone traits might be informative

Response: We have included LD score regression analysis across the Icelandic and UK Biobank samples between the DXA area, and height, BMD and the other bone traits in the manuscript.

p 5 GEFOS spelt incorrectly

Response: We have now corrected this mistake.

Will the authors be making the GWAS summary results data available to the public?

Response: The GWAS summary data are available upon request.

Reviewer #2 (Remarks to the Author):

Genome wide association study of bone size yields twelve loci that also affect height, bone density, osteoarthritis and fractures
Styrkarsdottir et al.

This article describes a GWAS in the Icelandic population of bone size at the hip, hip sub-areas, and the upper four lumbar vertebrae. Replication has taken place in a variety of cohorts, and attempts have been made to disentangle the relationship between bone size, height, BMD, and hip fractures and OA.

The work is novel, and will be of interest to the scientific community.

The paper overall is well-written, however it would benefit from some editing for clarity. In particular, the authors use bone area and bone size interchangeably, in the supplementary tables DXA size is used. One term should be used throughout. Furthermore, 16 variants at 14 loci were discovered, three failed to replicate (16-3=13), but the authors say that 12 loci were discovered – where this discrepancy lies is not obvious (this also relates to comment 4).

Response: We have now changed bone size to bone area at relevant places in the manuscript.

We report on 16 associations at 14 loci (2 at SOX9, and 2 at 15q25.2) in the discovery analysis, 3 of which did not replicate. One of those signals is at 15q25.2 which has two signals. The number of loci that replicate are thus 12.

Major Comments:

1. Sub-phenotyping, especially in terms of sub-regions of the hip, is critical to the results of this study, but the way in which the bone areas were derived is not described (it may be automated within the software). Sup Fig 1 allegedly shows how the different parameters were derived, but in fact fails to do so. In particular:

- a. what determines the placement of the dotted yellow line box that defines the femoral neck?**
- b. What determines the angle of the red solid line forming the border between the [greater] trochanter, and the “intertrochanteric” (this is not how the word is used clinically or anatomically) area?**

Small changes in placement of these lines can have large implications on the size measurement ie. the phenotype of the GWAS.

2. Was the calculation of bone areas performed automatically by image recognition software, or was each scan read manually by an individual? Please provide references for the software, including estimates of intra- and inter-observer reliability. Different systems used in the different cohorts will no doubt give different results – what is the “inter-machine” correlation of these important parameters?

Response to comments 1 and 2: Measurement of the different parameters is automated within the DXA machine software. The operating technicians, who are trained for this task, carefully monitor the

placement of the individual in the machine prior to measurement. They also indicate after the scanning if a particular measure is not valid for any reason. Each machine is calibrated every morning before operation using a phantom. The two DXA machines used in Iceland were further calibrated by measuring a group of the same individuals on both machines.

There are two main manufacturers of DXA machines, Hologic (<https://www.hologic.com/hologic-products/breast-skeletal/horizon-dxa-system>) and Lunar (<https://www.gehealthcare.com/en/products/bone-health-and-metabolic-health>), each with their own software and means to optimize the results. All the measures for the Icelandic discovery samples were done using Hologic machines whereas the replication studies used either Hologic or the Lunar machines. There are some differences between the two machines, which are difficult to fully account for. However, using standardized measures rather than absolute values from the machines we believe most of those differences are accounted for in our analyses.

Supplementary Figure 1 was borrowed from the manufacturer of the Hologic DXA machines and was solely meant as an illustration of the different areas measured. The lines are also only illustrative and determined automatically by image recognition software within each DXA machine.

The Hologic manufacturers refer to the area as intertrochanteric, the Lunar manufacturers as shaft. This includes also the lesser trochanter. We have now included shaft at the first mention of the intertrochanteric area (intertrochanteric/shaft).

We have now included the above information in the method section of the manuscript.

3. I note that the femoral head was excluded from analysis, but a major stream of analysis in the paper examines the overlap with hip OA – a disease formed between the femoral head and acetabulum. Why was the femoral head excluded? Can it be included?

Response: The femoral head is simply not included in the output from the DXA machines.

4. It is not explicitly stated how the “loci” at SH3GL3 and ADAMTSL3 were separated. Was this on the basis of LD, distance, or both? No conditional analysis has been performed (or at least reported) at any locus. It is not uncommon for there to be more than one independent signal at a particular locus, so the analysis should be repeated conditioning on the top SNP to look for residual association at each locus. For example, the locus at POLD3 appears to cross a recombination peak, and may represent two signals.

Response: Conditional analysis was performed for all loci. We did report two signals at 17q24.3 (SOX9) and at 15q25.2 (SH3GL3/ADAMTSL3) based on conditional analysis (Supplementary Table 2). Since there were no additional significant association signals to report from conditional analyses across the loci, we chose not to report on this analysis further. This should also be evident from Supplementary Figure 2, which shows locus zoom plots and LD between markers across the locus for all the associations.

We now indicate that we performed conditional analysis across the loci within the main text: “Sixteen variants, all common, at 14 loci satisfied our criteria of genome-wide significance³⁰ in the association

analysis, and after conditional analysis across the loci”

5. In table 1, three markers did not replicate. These should be relegated to the status of “suspected” loci rather than “confirmed”, and the table, title, and discussion should be adjusted to reflect this.

Response: These non-replicating loci are not included as confirmed loci (a wording that is not in the manuscript). We refer to table 1 -“Genome wide significant associations of DXA area”- when describing the significant results in the Icelandic discovery cohort. We do, however, include these three variants in analyses of association with other traits, showing strong association with height for two of them and FDR adjusted association with osteoarthritis for one of those three variants.

We have now changed the title of Table 1 to “Genome wide significant associations of DXA area in the Icelandic discovery samples, and replication results”.

6. It is suggested throughout the manuscript that the closest gene to the lead SNP is the one potentially responsible for the phenotype. There is a large literature on this often being untrue. It would be much better to perform a genome-wide gene-based test of association (eg MAGMA), and running gene-set enrichment analysis on the results – these analyses can be implemented in FUMA.

Response: We use the closest gene to name the associating loci. We did not intend to indicate that the closest gene is responsible for the phenotype. In Supplementary Table 7 we had included functional annotations of the associated variants in terms of chromatin marks indicative of gene promoters, enhancers and their predicted gene targets. The closest gene of course is sometimes the gene that functional annotations picks as the most likely gene. We have now added as a separate section in the Results “*Functional annotation of variants and enrichment analysis*” that describes tissue and gene set enrichment analysis using Depict (Pers et al, Nat Comm 2015) together with functional annotations.

7. Line 131-134. Here it is suggested that rs3753841 associates with reduced osteophytes in knee and hand. Checking Sup table 7 reveals p values for these two associations of 0.89 and 0.52. Please correct this.

Response: We thank the reviewer to point out this mistake and have removed the statement on osteophytes. This was an unfortunate editing error, the osteophyte association was originally meant as a discussion of another variant that we later removed from this discussion. We apologize for this mistake.

8. Line 140-142. Table 2. rs11614913 shows significant heterogeneity between studies when tested for association with hip fracture. Can we see a Forest plot to show this, and do the authors have any explanation why it may show opposite effects in different populations?

Response: We have now included a Forest plot for association of rs11614913 with hip fractures in the manuscript. All four sample-sets show the same direction of effects:

9. The associations between height, bone area, and BMD are indeed complex, and the authors have made efforts to disentangle this. Did they consider formal Mendelian Randomisation to look for a “causal” effect between the phenotypes?

Response:

Supplementary Figures 9 and 10 represent formal Mendelian Randomization analysis. They clearly show that there is not a simple “causal” relationship between height and BMD on the one hand and bone area on the other. We are not sure if the reviewer is referring to some specific other analysis.

Minor comments

Line 121 Extra closed bracket

Response: We have now corrected this mistake.

Line 180 grammatically incorrect

Response: We cannot see how the sentence in line 180 is grammatically incorrect. We have, however, now changed the sentence to: “Furthermore, after correcting for the number of markers tested additional 24 of the 697 reported height variants¹ (P Bonferroni $\leq 7.2 \times 10^{-5}$) associate with DXA area measures.”

Line 291 “OA” for “other allele” is mentioned in the legend but not shown in the table

Response: We have now corrected this mistake.

Line 460 missing reference

Response: The missing reference (bracket in the manuscript) referred to a website that is included under the URL section. We have now removed the parenthesis.

Line 487 the UK Biobank corrected imputation was released a couple of months ago – could this not have been incorporated?

Response: This manuscript was well under way when the correct UK Biobank imputation was released. None of the variants in the Icelandic discovery study were excluded from the UK Biobank because of the problematic imputation release.

Supp Fig 2 – please colour all variants considered statistically significant in the discovery cohorts in a different colour (eg red) to aid their identification.

Response: We have now included a line across the Manhattan plots that shows the strictest P-value cutoff of $7.9E-10$. This figure is now included in the main text.

Table 2 and 3 – please highlight in bold the significant associations to make the table easier to read.

Response: We have now indicated the significant associations by an asterisk.

Reviewer #3 (Remarks to the Author):

The authors present interesting GWAS data and they aim at functionally evaluating the top GWAS variant that associates with lumbar spine area and hip fracture. The GWAS data are novel. I have some major comments re the functional exploration of the rs11614913 lead SNP and its associated variants. I recommend additional experimentation in trait-relevant cell types as well as clarification of presented results with respect to the pathology/biology (hip fracture/lumbar spine density)

Major comments

1. The hypothesis that rs11614913 might affect the thermostability of mature miRNA-196a duplex through introducing the wobble bond in miRNA duplex and thus alter the efficiency of target gene repression is plausible (not entirely novel). Please see <https://doi.org/10.3390/ijms18122529> - how do the authors comment on the results presented in this paper?

Response: We thank the reviewer for pointing out this paper, which proposes that rs11614913 affects abundance of miR-196a-5p through processing of the precursor miR and have included this possibility in the main text. The experiments that we performed cannot discriminate between effects of rs11614913 on stability or processing of miR-196a-5p.

Can the authors investigate the proposed candidate mechanism in more detail? E.g. testing efficiency of Ago loading of miRNA-196 on target suppression by luciferase reporter gene assays (co-transfection of miRNA_wt/mut overexpression plasmids with reporter vectors containing miRNA recognition sites); immunoprecipitation experiments with various flag-tagged human Ago proteins with subsequent detection of immuno-precipitated guide and passenger miRNAs in single stranded or duplex miRNA forms would give further insight, etc.

Response: These are interesting suggestions, but require extensive follow-up studies to gain a better understanding of the molecular mechanism by which rs11614913 affects the miR-196a-5p function that are beyond the scope of this paper.

2. RNAseq in HEK cells (Fig. 1., Suppl Fig 5): It is not clear how the differences in the suppression of the identified miRNA target genes (by RNAseq in HEK cells) could functionally associate with hip fracture and lumbar spine bone density. Can the authors comment on the functionality of the target genes with respect to these traits? Are these genes expressed and affected by the genotype also in trait-relevant cells e.g. osteoblasts?

Response: For clarification we have now added to the manuscript analysis of the biological function of the seventeen high-confident miR-196a-5p target based on gene ontology (Supplementary Table 10) According to this analysis, genes, mostly transcription factors, involved in anterior/posterior pattern specification were highly overrepresented (25-fold; $P=1.7e-5$) in addition to skeletal system development (13-fold; $P\text{-value}=2.5e-5$).

We additionally show information on expression in osteoblasts obtained from the Fantom5 dataset, this is listed in the same Supplementary Table. Unfortunately, we do not have access to a series of samples derived from osteoblasts from enough individuals that would enable us to explore the impact of rs11614913 on gene expression in trait relevant cells.

Definition of a group of high-confidence miR-196a-5p targets is not very convincing. It is not clear to me why the Pct score of exactly >0.86 was used. Did this score provide best results when referring to RNAseq analysis and data presentation (Fig. 1 and Suppl. Fig. 5.)? Can the RNA-seq data be analysed taking into account the miR-196a-5p targets that have been supported experimentally using the gold standard reporter gene assays (e.g. see DIANA-TarBase)?

Response: To define the Pct threshold, we set out to determine the lowest Pct threshold at which the fraction of genes with experimental support (direct interactions to miR-196a-5p; Tarbase V8) above the threshold is maximized. This was carried out in incremental steps of 0.02 (Pct from 0.42 to 0.94). For completeness, we have now added sentences describing this procedure in the Supplementary note, along with a Supplementary figure (included below) that clearly shows the sharp increase in the fraction of genes with experimental support at $Pct > 0.86$. The experimentally determined direct interactions between miR and mRNAs can only be regarded as suggestive of functional consequences as, for example, weak or transient interactions are likely irrelevant and interactions detected are dependent upon expression of the target mRNAs. For these reasons, we choose to use *in silico* predictions as the basis for defining target genes and, as evolutionarily conserved sites are more likely functionally relevant, we

therefore specifically looked at the Pct score. We reasoned that experimentally supported interactions from Tarbase would be helpful to determine the threshold for the Pct score.

Regarding the latter part of the reviewer's question, the six genes with positive direct interaction with miR-196a-5p derived from the reporter gene assay are already contained in the set of genes used to determine the optimal Pct threshold. Thus, we are already accounting for these six genes out of the total of 1174 genes from Tarbase annotated as direct targets of miR-196a-5p. The reporter assay is a candidate gene approach and, as such, only a few genes have currently been tested with respect to miR-196a-5p which severely limits the usability of data derived from these assays.

Several other transcripts seem altered by the overexpression of the variant-containing miRNA sequences in HEK cells. What are these targets?

Response: Genes found down-regulated between MIR196A2 insert versus control empty plasmid cells reflect 1) primary effects and 2) secondary effects. The primary effects are the direct target genes of miR-196a-5p whereas the secondary effects are downstream effects caused by downregulation of the target genes, e.g. many of the high-confidence miR-196a-5p target genes are transcription factors such as the homeobox proteins HOXA7, HOXC8, HOXA9 and others including ZBTB26. Indeed, the statistical overrepresentation test (mentioned in response to a previous question from this reviewer) carried out on the list of high confidence target genes detected a 3-fold overrepresentation of genes involved in regulation of expression within the list of high-confidence target genes (P-value=6.7e-6).

In order to study the biological functions of the entire set of genes found differentially expressed between MIR196A2 insert and empty vector plasmid controls, i.e. both primary and secondary effects, we carried out a gene set enrichment analysis. This analysis identifies enrichment for energy metabolism, growth factor response and embryonic skeletal system morphogenesis. This is now included in the Supplementary note and Supplementary Figure 7.

Taking 14 targets among these e.g. downregulated genes could yield similar plots as in Fig 1, Suppl. Fig 5 (based on predicted miR-196-5p mRNA targets) etc.

Response: We emphasize that we did not make use of the RNAseq data to define our list of miR-196a-5p target genes. As pointed out in response to the question above, a substantial fraction of the differentially expressed genes are likely secondary downstream effects resulting from miR-196a-5p knock-down of genes encoding transcription factors which will then clearly affect multiple other genes (i.e. genes not directly targeted by miR-196a-5p). Thus, the differential expression analysis is complicated by presence of both primary and secondary effects. The conservative approach that we took was to obtain a high-quality list of direct miR-196a-5p target genes, i.e. primary effects, independently of the differential expression analysis and we then carried out a permutation test to assess whether the log₂ fold change values derived from the C- versus T-allele comparison were directionally biased.

3. Other mechanisms beyond the effect of the genotype on miRNA 196-5p biology are possible. How were the lead SNP and its associated variants prioritized as most probably causal for the association with the trait, in particular for the rs11614913. The functional annotations, particularly for the rs11614913 and its associated variants, should be discussed beyond descriptive reporting in Suppl. Table 6, specifically in terms of potential functional variant prioritization. E.g, it could be that the promoter-located rs56154542 (the associated variant) rather than rs11614913 (the lead) determines HOXC8 levels. Can the authors test the associations of rs56154542 variation with HOXC8 expression? It is not clear to me why HOXC8 levels are reported in adipose tissue - the SNP effects can be cell-type specific and trait-relevant cells (e.g. osteoblasts) should be used to test the associations of the genotype with target gene expression (HOXC8 and other targets as determined by RNA-seq screen in HEK cells). Which of the affected targets in HEK cells (as defined by RNAseq – Fig 1) are expressed in e.g. osteoblasts? Do any of the miR-196-5p targets affect osteoblast biology.

Response: We have now expanded the results section on functional annotation of the non-coding variants. The rs11614913 variant has the most significant P-value and we note that a correlated variant is located in the HOXC8 promoter. We do not claim that the association with area is only mediated through rs11614913 subtle, yet significant, effect on miR-196a-5p target genes in the HEK293T overexpression experiments. It can be added here that PICS (probabilistic identification of causal SNPs), a fine-mapping algorithm developed by Farh et al (Nature, 2015), identifies rs11614913 as the sole candidate causal variant at this locus (see table below).

PICS (probabilistic identification of causal SNPs) analysis for rs11614913 to determine the most likely causal SNP at this locus (see online algorithm at pubs.broadinstitute.org/pubs/finemapping/pics.php)

Index_SNP	Linked_SNP	Dprime	Rsquare	Phase	PICS_probability
rs11614913	rs11614913	1	1	N,N	0.9892
rs11614913	rs3803042	0.9887	0.8966	C,G	0.0071
rs11614913	rs56154542	0.9883	0.8606	C,G	0.0018
rs11614913	rs201281812	0.9499	0.8548	C,R	0.0014
rs11614913	rs138542768	0.9124	0.8191	C,R	0.0003
rs11614913	rs199695387	0.9466	0.7348	C,R	0
rs11614913	rs3840780	0.8751	0.6598	C,R	0
rs11614913	chr12:54410013	0.8887	0.6394	C,R	0
rs11614913	rs754133	0.8342	0.5699	C,G	0
rs11614913	rs894738	0.8336	0.5658	C,G	0
rs11614913	rs894737	0.8336	0.5658	C,A	0
rs11614913	rs12426399	0.8336	0.5658	C,G	0
rs11614913	rs4759318	0.8330	0.5618	C,C	0
rs11614913	rs12319419	0.8219	0.5532	C,G	0
rs11614913	rs10876528	0.8009	0.5404	C,C	0
rs11614913	rs10747689	0.7949	0.5323	C,C	0
rs11614913	rs61921797	0.7942	0.5283	C,G	0
rs11614913	rs2071449	0.7942	0.5283	C,C	0
rs11614913	rs12422600	0.7942	0.5283	C,G	0
rs11614913	rs4759319	0.7896	0.5282	C,G	0
rs11614913	rs4759316	0.9852	0.5208	C,G	0
rs11614913	rs1109391	0.9852	0.5208	C,G	0

The rs56154542 variant was mistakenly included as a promoter-variant for HOXC8, it is a promoter variant for HOXC9. We have revised the table and we apologize for the mistake. However, the rs371683123 variant (chr12:54009024(indel)), a 3 bp deletion, was correctly stated as a HOXC8 promoter variant. We now more fully describe the potential contribution of other variants found highly correlated with rs11614913.

We tested the correlation of the area associated variants with HOXC8 expression in adipose tissue as we have access to comprehensive RNA sequencing data from adipose tissue (from 746 individuals) together with whole genome sequence information. Similar dataset for osteoblasts was not available to us. We have approached the authors of Grundberg et al (PMID:19654370), who describe expression chip data and SNP chip data in osteoblasts from 95 individuals, however, without luck. We are not aware of any other potential dataset to use for such an analysis. The GEO accession for the Grundberg et.al. data (GSE15678) only includes expression levels and not genotypes for correlation analyses.

Adipocytes and osteoblasts share origin from mesenchymal stem cells, however, we are fully aware that the regulatory mechanism between these cell types may be different. We also note that bone is composed of other cell types that may be involved.

Minor Comments:

- In comparison to empty control, T allele vector does not repress GAN while C allele vector does (rs11614913 lead SNP Suppl. Table 10). Can the authors comment on this?

Response: This result nicely reflects our interpretation that the T-allele is less effective in repressing miR-196a-5p target genes. The reviewer is referring to a particular gene, *GAN*, being down-regulated in C-allele, and not in T-allele. The direct gene-by-gene comparison between C-allele and T-allele, however, gives a non-significant result. This is because the effect sizes in this comparison are subtle and, as a result, we were unable to demonstrate biological effect on the gene-by-gene basis. As the subtle effect sizes leave us underpowered for the gene-by-gene approach, we set out to assess whether significant effects emerge by analyzing the data across genes, i.e. across the defined direct target genes. And, we find that, as a group, our defined list of target genes are significantly down-regulated in the C-allele (as compared with T-allele), but as we are unable to determine this on a gene-by-gene basis we cannot single out a particular gene.

- Please refer to lumbar spine (not spine only) are in abstract and text...

Response: We now consistently refer to lumbar spine through the manuscript

- rs11614913 lead SNP was shown to affect the expression levels of miR-196-5p (e.g. doi: 10.1038/srep19825) in seminal plasma. It would be important to measure the effects of the genotype on the expression of miR-196-5p in cells relevant for hip fracture/lumbar spine area traits, e.g. osteoblasts, if possible. Is the expression of miR-196-3p also affected?

Response: We agree, it is important to investigate the effect of rs11614913 on expression in osteoblasts, or other bone or cartilage cells, however, we do not have access to such data as previously stated.

Reviewer #1 (Remarks to the Author):

The authors have satisfactorily addressed my comments.

However, I'm not particularly happy with their statement that the GWAS summary results are available on request from them. In my opinion this is against the spirit of open and transparent science and frankly open to abuse. It costs very little to post summary GWAS on a publicly available website- like the rest of the GWAS community do- and DECODE should not get a free pass in this respect.

Reviewer #2 (Remarks to the Author):

Reviewer #2 (Remarks to the Author):

Genome wide association study of bone size yields twelve loci that also affect height, bone density, osteoarthritis and fractures
Styrkarsdottir et al.

Review of first revision

The paper has improved, and the analysis is more thorough. However, there remain a couple of areas of concern that have not been addressed: clarity over the 13 replicated signals, and MR analysis – see below for detailed comments.

One point that was raised by another reviewer is data availability. It would be better for the summary stats to be deposited in repository rather than “upon request”. This limited level of availability is not in the spirit of data sharing that Nature Communications demands.

Furthermore, 16 variants at 14 loci were discovered, three failed to replicate (16-3=13), but the authors say that 12 loci were discovered – where this discrepancy lies is not obvious (this also relates to comment 4).

Response: We have now changed bone size to bone area at relevant places in the manuscript.
Reviewer's reply: DXA area still persists in the abstract, tables 2 and 3, etc. please be consistent or it will confuse the readership. Maybe it is best to use DXA area, as this more accurately represents the methods used.

We report on 16 associations at 14 loci (2 at SOX9, and 2 at 15q25.2) in the discovery analysis, 3 of which did not replicate. One of those signals is at 15q25.2 which has two signals. The number of loci that replicate are thus 12.

Reviewer's reply: Please would the authors clarify this throughout the manuscript. For example, in the abstract they could write:

We found thirteen independent association signals at twelve loci that replicate in samples of European and East Asian descent (N = 13,608 – 21,277).

See also point 5.

The title also remains confusing from this point of view, and should also be changed for clarity. The title also suggests that all loci affect all of the other traits. This is not the case, so the title is misleading.

Major Comments:

1. Sub-phenotyping, especially in terms of sub-regions of the hip, is critical to the results of this study, but the way in which the bone areas were derived is not described (it may be automated within the software). Sup Fig 1 allegedly shows how the different parameters were derived, but in fact fails to do so. In particular:

- a. what determines the placement of the dotted yellow line box that defines the femoral neck?
- b. What determines the angle of the red solid line forming the border between the [greater] trochanter, and the "intertrochanteric" (this is not how the word is used clinically or anatomically) area?

Small changes in placement of these lines can have large implications on the size measurement ie. the phenotype of the GWAS.

2. Was the calculation of bone areas performed automatically by image recognition software, or was each scan read manually by an individual? Please provide references for the software, including estimates of intra- and inter-observer reliability. Different systems used in the different cohorts will no doubt give different results – what is the "inter-machine" correlation of these important parameters?

Response to comments 1 and 2: Measurement of the different parameters is automated within the DXA machine software. The operating technicians, who are trained for this task, carefully monitor the placement of the individual in the machine prior to measurement. They also indicate after the scanning if a particular measure is not valid for any reason. Each machine is calibrated every morning before operation using a phantom. The two DXA machines used in Iceland were further calibrated by measuring a group of the same individuals on both machines.

There are two main manufacturers of DXA machines, Hologic (<https://www.hologic.com/hologic-products/breast-skeletal/horizon-dxa-system>) and Lunar (<https://www.gehealthcare.com/en/products/bone-health-and-metabolic-health>), each with their own software and means to optimize the results. All the measures for the Icelandic discovery samples were done using Hologic machines whereas the replication studies used either Hologic or the Lunar machines. There are some differences between the two machines, which are difficult to fully account for. However, using standardized measures rather than absolute values from the machines we believe most of those differences are accounted for in our analyses.

Supplementary Figure 1 was borrowed from the manufacturer of the Hologic DXA machines and was solely meant as an illustration of the different areas measured. The lines are also only illustrative and determined automatically by image recognition software within each DXA machine. The Hologic manufacturers refer to the area as intertrochanteric, the Lunar manufacturers as shaft. This includes also the lesser trochanter. We have now included shaft at the first mention of the intertrochanteric area (intertrochanteric/shaft).

We have now included the above information in the method section of the manuscript.

Reviewer's Reply: Thank-you for clarification.

3. I note that the femoral head was excluded from analysis, but a major stream of analysis in the paper examines the overlap with hip OA – a disease formed between the femoral head and acetabulum. Why was the femoral head excluded? Can it be included?

Response: The femoral head is simply not included in the output from the DXA machines.

Reviewer's Reply: Please note this in discussion as a limitation of the study

4. It is not explicitly stated how the "loci" at SH3GL3 and ADAMTSL3 were separated. Was this on the basis of LD, distance, or both? No conditional analysis has been performed (or at least reported) at any locus. It is not uncommon for there to be more than one independent signal at a particular locus, so the analysis should be repeated conditioning on the top SNP to look for residual association at each locus. For example, the locus at POLD3 appears to cross a recombination peak, and may represent two signals.

Response: Conditional analysis was performed for all loci. We did report two signals at 17q24.3 (SOX9) and at 15q25.2 (SH3GL3/ADAMTSL3) based on conditional analysis (Supplementary Table 2). Since there were no additional significant association signals to report from conditional analyses across the loci, we chose not to report on this analysis further. This should also be evident from Supplementary Figure 2, which shows locus zoom plots and LD between markers across the locus for all the associations.

We now indicate that we performed conditional analysis across the loci within the main text:

"Sixteen variants, all common, at 14 loci satisfied our criteria of genome-wide significance³⁰ in the

association analysis, and after conditional analysis across the loci”

Reviewer’s Reply: OK

5. In table 1, three markers did not replicate. These should be relegated to the status of “suspected” loci rather than “confirmed”, and the table, title, and discussion should be adjusted to reflect this.

Response: These non-replicating loci are not included as confirmed loci (a wording that is not in the manuscript). We refer to table 1 -“Genome wide significant associations of DXA area”- when describing the significant results in the Icelandic discovery cohort. We do, however, include these three variants in analyses of association with other traits, showing strong association with height for two of them and FDR adjusted association with osteoarthritis for one of those three variants. We have now changed the title of Table 1 to “Genome wide significant associations of DXA area in the Icelandic discovery samples, and replication results”.

Reviewer’s Reply: I think this needs attention in the manuscript, not just the table title. For example, in the paragraph beginning “We followed up these sixteen...”, the authors could add that the three signals that didn’t replicate are possibly false positive associations. I think that these three variants should be removed from downstream analyses in the paper.

6. It is suggested throughout the manuscript that the closest gene to the lead SNP is the one potentially responsible for the phenotype. There is a large literature on this often being untrue. It would be much better to perform a genome-wide gene-based test of association (eg MAGMA), and running gene-set enrichment analysis on the results – these analyses can be implemented in FUMA.

Response: We use the closest gene to name the associating loci. We did not intend to indicate that the closest gene is responsible for the phenotype. In Supplementary Table 13 we had included functional annotations of the associated variants in terms of chromatin marks indicative of gene promoters, enhancers and their predicted gene targets. The closest gene of course is sometimes the gene that functional annotations picks as the most likely gene. We have now added as a separate section in the Results “Functional annotation of variants and enrichment analysis” that describes tissue and gene set enrichment analysis using Depict (Pers et al, Nat Comm 2015) together with functional annotations.

Reviewer’s Reply: You may not intend to indicate that the closest gene is responsible for the phenotype, but this is how it reads. I agree that sometimes the closest gene is the most likely candidate. In table 1, it would probably be best to re-label the column “gene” as “closest gene” – are these actually the closest genes always? If not, perhaps “candidate gene” might be better.

7. Line 131-134. Here it is suggested that rs3753841 associates with reduced osteophytes in knee and hand. Checking Sup table 7 reveals p values for these two associations of 0.89 and 0.52. Please correct this.

Response: We thank the reviewer to point out this mistake and have removed the statement on osteophytes. This was an unfortunate editing error, the osteophyte association was originally meant as a discussion of another variant that we later removed from this discussion. We apologize for this mistake.

Reviewer’s Reply: OK

8. Line 140-142. Table 2. rs11614913 shows significant heterogeneity between studies when tested for association with hip fracture. Can we see a Forest plot to show this, and do the authors have any explanation why it may show opposite effects in different populations?

Response: We have now included a Forest plot for association of rs11614913 with hip fractures in the manuscript. All four sample-sets show the same direction of effects:

Reviewer’s Reply: Thank-you for including this data.

9. The associations between height, bone area, and BMD are indeed complex, and the authors have made efforts to disentangle this. Did they consider formal Mendelian Randomisation to look for a "causal" effect between the phenotypes?

Response:

Supplementary Figures 9 and 10 represent formal Mendelian Randomization analysis. They clearly show that there is not a simple "causal" relationship between height and BMD on the one hand and bone area on the other. We are not sure if the reviewer is referring to some specific other analysis.

Reviewer's Reply: I cannot see any MR analysis. These analyses are simple correlations between the effects of SNPs in GIANT or GEFOS and the measures used in this paper.

In order to do proper MR, the authors should use a two sample MR approach implemented in the MendelianRandomisation package for R (or equivalent), using the GIANT or GEFOS SNPs as instrumental variables.

The package will calculate Wald Statistics for each SNP, and then allow you to perform MR using different methods - I would suggest Inverse Variant Weighted MR as the simplest, followed by MR-Egger regression to look for horizontal pleiotropy, and finally weighted median (which is more robust to violation of the InSIDE assumption than MR-Egger). These analyses will provide more robust evaluation of the "causal" relationship between the parameters.

Please see: Zengini et al Nat Genet. 2018 Apr;50(4):549-558 for an example of well performed MR - I note that many authors on that paper are authors on this paper.

Minor comments

Reviewer's Reply: All now OK.

Reviewer #3 (Remarks to the Author):

The authors addressed many of my comments appropriately. Some true functional data would still be desirable. I do not see these experiments as entirely out of the scope of current work. In any case, the authors should clearly mention the limitations of their study including lack of true functional data in trait relevant cells.

Minor comment:

- there is a mistake in SNP number in the title of the Suppl Fig. 8
- the resolution of Suppl Fig 7 should be improved

Reviewer #1 (Remarks to the Author):

The authors have satisfactorily addressed my comments.

However, I'm not particularly happy with their statement that the GWAS summary results are available on request from them. In my opinion this is against the spirit of open and transparent science and frankly open to abuse. It costs very little to post summary GWAS on a publicly available website-like the rest of the GWAS community do- and DECODE should not get a free pass in this respect.

Response: The GWAS summary statistics will be made publicly available.

In our opinion, we can primarily thank the large international collaboration consortia for the spirit of open and transparent science in the GWA community, not whether the summary results from individual studies are public or not.

Reviewer #2 (Remarks to the Author):

The paper has improved, and the analysis is more thorough. However, there remain a couple of areas of concern that have not been addressed: clarity over the 13 replicated signals, and MR analysis – see below for detailed comments.

One point that was raised by another reviewer is data availability. It would be better for the summary stats to be deposited in repository rather than “upon request”. This limited level of availability is not in the spirit of data sharing that Nature Communications demands.

Response: The GWAS summary statistics will be made publicly available.

Furthermore, 16 variants at 14 loci were discovered, three failed to replicate (16-3=13), but the authors say that 12 loci were discovered – where this discrepancy lies is not obvious (this also relates to comment 4).

Response: We have now changed bone size to bone area at relevant places in the manuscript.

Reviewer's reply: DXA area still persists in the abstract, tables 2 and 3, etc. please be consistent or it will confuse the readership. Maybe it is best to use DXA area, as this more accurately represents the methods used.

Response: The previous critique related to the use of bone size and bone area, not the use of DXA area versus bone area. We have now changed DXA area to DXA bone area at relevant places in the manuscript.

We report on 16 associations at 14 loci (2 at SOX9, and 2 at 15q25.2) in the discovery analysis, 3 of

which did not replicate. One of those signals is at 15q25.2 which has two signals. The number of loci that replicate are thus 12.

Reviewer's reply: Please would the authors clarify this throughout the manuscript. For example, in the abstract they could write:

We found thirteen independent association signals at twelve loci that replicate in samples of European and East Asian descent (N = 13,608 – 21,277).

See also point 5.

Response: We have now changed the abstract to include this sentence.

The title also remains confusing from this point of view, and should also be changed for clarity. The title also suggests that all loci affect all of the other traits. This is not the case, so the title is misleading.

Response: We have changed the title to read: "Genome wide association study of bone size yields twelve loci that also affect height, bone density, osteoarthritis or fractures", as all but one of the variants associate with any of these traits/diseases.

Major Comments:

3. I note that the femoral head was excluded from analysis, but a major stream of analysis in the paper examines the overlap with hip OA – a disease formed between the femoral head and acetabulum.

Why was the femoral head excluded? Can it be included?

Response: The femoral head is simply not included in the output from the DXA machines.

Reviewer's Reply: Please note this in discussion as a limitation of the study

Response: We now state that the femoral head is not included in the output from the DXA machines.

5. In table 1, three markers did not replicate. These should be relegated to the status of "suspected" loci rather than "confirmed", and the table, title, and discussion should be adjusted to reflect this.

Response: These non-replicating loci are not included as confirmed loci (a wording that is not in the manuscript). We refer to table 1 - "Genome wide significant associations of DXA area"- when

describing the significant results in the Icelandic discovery cohort. We do, however, include these three variants in analyses of association with other traits, showing strong association with height for two of them and FDR adjusted association with osteoarthritis for one of those three variants.

We have now changed the title of Table 1 to "Genome wide significant associations of DXA area in the Icelandic discovery samples, and replication results".

Reviewer's Reply: I think this needs attention in the manuscript, not just the table title. For example, in the paragraph beginning "We followed up these sixteen....", the authors could add that the three

signals that didn't replicate are possibly false positive associations. I think that these three variants should be removed from downstream analyses in the paper.

Response: We think it is informative to include these three variants in the downstream analysis as two of them also associate significantly with height and one with hip osteoarthritis.

6. It is suggested throughout the manuscript that the closest gene to the lead SNP is the one potentially responsible for the phenotype. There is a large literature on this often being untrue. It would be much better to perform a genome-wide gene-based test of association (eg MAGMA), and running gene-set enrichment analysis on the results – these analyses can be implemented in FUMA.

Response: We use the closest gene to name the associating loci. We did not intend to indicate that the closest gene is responsible for the phenotype. In Supplementary Table 13 we had included functional annotations of the associated variants in terms of chromatin marks indicative of gene promoters, enhancers and their predicted gene targets. The closest gene of course is sometimes the gene that functional annotations picks as the most likely gene. We have now added as a separate section in the Results "Functional annotation of variants and enrichment analysis" that describes tissue and gene set enrichment analysis using Depict (Pers et al, Nat Comm 2015) together with functional annotations.

Reviewer's Reply: You may not intend to indicate that the closest gene is responsible for the phenotype, but this is how it reads. I agree that sometimes the closest gene is the most likely candidate. In table 1, it would probably be best to re-label the column "gene" as "closest gene" – are these actually the closest genes always? If not, perhaps "candidate gene" might be better.

Response: We already stated in the table legend that "Gene refers to the nearest gene". For clarification we have changed the "Gene" column to "Closest gene".

9. The associations between height, bone area, and BMD are indeed complex, and the authors have made efforts to disentangle this. Did they consider formal Mendelian Randomisation to look for a "causal" effect between the phenotypes?

Response:

Supplementary Figures 9 and 10 represent formal Mendelian Randomization analysis. They clearly show that there is not a simple "causal" relationship between height and BMD on the one hand and bone area on the other. We are not sure if the reviewer is referring to some specific other analysis.

Reviewer's Reply: I cannot see any MR analysis. These analyses are simple correlations between the effects of SNPs in GIANT or GEFOS and the measures used in this paper.

In order to do proper MR, the authors should use a two sample MR approach implemented in the MendelianRandomisation package for R (or equivalent), using the GIANT or GEFOS SNPs as instrumental variables.

The package will calculate Wald Statistics for each SNP, and then allow you to perform MR using different methods - I would suggest Inverse Variant Weighted MR as the simplest, followed by MR-Egger regression to look for horizontal pleiotropy, and finally weighted median (which is more robust to violation of the InSIDE assumption than MR-Egger). These analyses will provide more robust evaluation of the "causal" relationship between the parameters.

Please see: Zengini et al Nat Genet. 2018 Apr;50(4):549-558 for an example of well performed MR – I note that many authors on that paper are authors on this paper.

Response: We have now performed the requested MR analysis with the following results (MR-Egger):

	Exposure	Estimate	Std Error	95% CI	P-value
FN BMD	Total hip area	0.059	0.105	-0.148, 0.265	0.58
	Intertrochanteric area	0.006	0.107	-0.204, 0.216	0.96
	Trochanter area	0.035	0.093	-0.147, 0.217	0.71
	Femoral neck area	0.249	0.107	0.040, 0.459	0.019
LS BMD	Lumbar spine area	-0.307	0.107	-0.518, -0.097	0.0040
Height	Total hip area	-0.065	0.052	-0.167, 0.037	0.21
	Intertrochanteric area	-0.047	0.045	-0.136, 0.041	0.30
	Trochanter area	-0.052	0.048	-0.146, 0.042	0.28
	Femoral neck area	-0.012	0.046	-0.102, 0.079	0.80
	Lumbar spine area	-0.102	0.055	-0.209, 0.006	0.064

We now include these results in a Supplementary Table 8.

Reviewer #3 (Remarks to the Author):

The authors addressed many of my comments appropriately. Some true functional data would still be desirable. I do not see these experiments as entirely out of the scope of current work. In any case, the authors should clearly mention the limitations of their study including lack of true functional data in trait relevant cells.

Response: We now include at the end of the discussion: "Together these data warrants further studies in bone relevant cells".

Minor comment:

- there is a mistake in SNP number in the title of the Suppl Fig. 8
- the resolution of Suppl Fig 7 should be improved

Response: We have now corrected the mistake in Suppl Fig. 8 and provided Suppl Fig 7 in higher resolution